# How does nitrogen control soil organic matter turnover and composition? - Theory and model

Chun Chung Yeung<sup>1</sup>, Harald Bugmann<sup>1</sup>, Frank Hagedorn<sup>2</sup>, Margaux Moreno Duborgel<sup>2</sup>, Olalla Díaz-Yáñez<sup>1</sup>

<sup>1</sup>Forest Ecology, Institute of Terrestrial Ecosystems, Department of Environmental Systems Science, ETH Zurich, 8092 Zurich, Switzerland

<sup>2</sup>Forest Soils and Biogeochemistry, Swiss Federal Institute for Forest, Snow and Landscape Research (WSL), Birmensdorf, Switzerland

Correspondence to: Chung C. Yeung (cyeung@ethz.ch)

10

**Abstract.** Nitrogen (N) enrichment triggers diverse responses of different soil organic carbon (SOC) pools, but a coherent mechanism to explain them is still lacking. To address this, we formulated several hypothesized N-induced decomposer responses in a dynamic soil model (irrespective of plant responses), i.e., decomposition retardation under increasing N excess and stimulation under decreasing N-limitation, N-responsive microbial turnover and carbon use efficiency (CUE), and a priming effect driven by changing microbial biomass. To evaluate their relevance on SOC turnover, they were incrementally combined into multiple model variants, and systematically tested against diverse data from meta-analyses of N addition experiments and SOC fraction data from contemporary temperate forests spanning wide environmental gradients.

Our results support the hypothesis that N directly influences multiple C pools by changing decomposition and microbial physiology, which are in turn driven by stoichiometric imbalances. Under N addition, only the model variants that incorporated both the responses of 1) decomposition retardation with increasing N-excess and 2) decomposition stimulation with decreasing N limitation were able to qualitatively reproduce the common observation of a greater increase of surface organic layer (LFH) relative to topsoil SOC, and of particulate organic carbon (POC) relative to mineral-associated carbon (MAOC). We attributed this to the accelerated decomposition of N-limited detritus by N addition, thereby supplying processed C to intermediate pools (i.e., POC and FH horizon). In addition, N retarded the decomposition of these processed C and MAOC that have lower C:N ratios. This concurrently explains the organic horizon and POC accumulation under contemporary N deposition in temperate forests, albeit with smaller effect sizes than in N addition experiments where N application rates are higher.

Furthermore, incorporating N-responsive microbial turnover and CUE helped reproduce microbial biomass reduction, and improved the modelling of microbial biomass C:N homeostasis and hence, the estimation of microbial N-limitation and excess. Collectively, our model experiment provided robust mechanistic insights on the stoichiometric control of soil N-C interaction.

We recommend our simple model for further testing and incorporation into other soil CN models.

Keywords: nitrogen addition; nitrogen deposition; soil organic carbon; process-based model; labile carbon and nitrogen; threshold elemental ratio; carbon use efficiency

## 1 Introduction

Forest soils are large reservoirs of soil organic carbon (SOC) and nitrogen (N) globally. However, the storage of C and N in forest soils varies greatly at the regional and local scale (van der Voort et al., 2016), reflected by varying organic matter (OM) composition across environmental gradients (i.e., the amount and proportion of deadwood, litter at varying decay stages, particulate organic matter, mineral-associated organic matter, etc.). Capturing this variation is a challenge as many factors drive OM composition, and soil models were typically only used to predict bulk SOC data (Le Noë et al., 2023; Zhang et al., 2020). As a result, soil models have repeatedly produced diverged, uncertain SOC projections under global change drivers (Bruni et al., 2022; Palosuo et al., 2012; Sulman et al., 2018; Todd-Brown et al., 2013; Wieder et al., 2018), likely because individual C pools (and their decomposers) respond differently to global change factors (Georgiou et al., 2024; Lugato et al., 2021; Rocci et al., 2021; Xiao et al., 2018). Constraining the proportions of C pools and their individual response are therefore essential to increase confidence in projections of future soil C balances.

Enhanced N deposition is a major global change factor, with up to 10-fold increase over the last century (Galloway et al., 2008). Recent meta-analyses have documented a general increase of bulk SOC stocks and especially surface organic horizon (LFH) under N deposition, where the accumulation increased with increasing dose and duration of N addition (Janssens et al., 2010; Liu & Greaver, 2010; Ramirez et al., 2012; Tang et al., 2023; Xu et al., 2021). However, this "general increase" is complicated by considerable variation among studies differing in location, form of N input, and initial OM composition whereby individual C pools respond variably (Jian et al., 2016; Tan et al., 2020; Tang et al., 2023; Waldrop et al., 2004; Zuccarini et al., 2023). Efforts to synthesize the diverse responses into a coherent theory and model are lacking, with soil N-C interaction being considered as a largely unresolved challenge to date (Chen et al., 2020; Cheng et al., 2018; Eastman et al., 2024; Hagedorn et al., 2012a; Sutton et al., 2008; Tang et al., 2023).

Experimental studies have since revealed key processes underlying the effects of N on OM dynamics. These include changes in plant litter production and allocation (Chen et al., 2015; Magnani et al., 2007), as well as microbial exo-enzyme activities (Chen et al., 2018a, b; Jian et al., 2016), biomass and physiology (Treseder, 2008; Yang et al., 2022; Zhang et al., 2018) — the focus of this study. Mechanistically, excess N suppresses the decomposition of some lignin-containing substrates (e.g., lignin enriched in particulate organic carbon – POC; Cotrufo & Lavallee, 2022), as well as protein-rich, low C:N compounds typical for mineral-associated organic matter (MAOC; Kopittke et al., 2018). This is related to the general suppression of both oxidative lignin enzyme and hydrolytic peptidase activities under excess N (Allison et al., 2008; Carreiro et al., 2000; Chen et al., 2018b; Gallo et al., 2004; Geisseler et al., 2010; Jian et al., 2016; Liu et al., 2023a; Rappe-George et al., 2017). In contrast, N was also found to stimulate the decomposition of labile C substrates (e.g., cellulose) and substrates with a very high C:N ratio (e.g., deadwood). This is due to the stimulation of cellulolytic enzyme activities (e.g., β-1,4-glucosidase) and the alleviation of N-limitation to microbial growth (Allison et al., 2009; Bebber et al., 2011; Carreiro et al., 2000; Hobbie et al.,

2012; Jian et al., 2016; Jing et al., 2021; Micks et al., 2004). Underlying these observed changes in decomposition are diverse decomposer responses, from physiology to community biomass and composition.

These decomposer responses have not been incorporated into CMIP6 soil models (Arora et al., 2020; Bouskill et al., 2014; Chen et al., 2019; Sun et al., 2016; Ťupek et al., 2016; Zaehle and Friend, 2010), despite the conclusion of multiple experimental studies that N addition exerted a larger effect on microbial decomposition than plant production. This is evidenced by decreased heterotrophic CO<sub>2</sub> fluxes despite increased litter C inputs, and the predominant preservation of "old" C instead of a gain of "new" C in isotopic studies (Bowden et al., 2019; Franklin et al., 2003; Frey et al., 2014; Griepentrog et al., 2015; Hagedorn et al., 2003; Liu et al., 2024). Eastman et al. (2024) recently investigated the CASA (one of the soil models in CMIP6) and the microbial-explicit MIMICS models under N addition. The models reproduced most plant but only limited soil responses, despite including a post-hoc parameter adjustment representing decomposition retardation. The authors attributed the mismatch to a lack of relevant N-responsive microbial processes. A few modelling studies have also investigated decomposer response to N addition (Chen et al., 2019; Tonitto et al., 2014; Whittinghill et al., 2012), but they again mainly featured a simple, static decomposition retardation factor on lignin decomposition, without considering other possible responses such as decomposition stimulation and microbial physiological changes dynamically (Fierer et al., 2012; Gravuer and Eskelinen, 2017; Hu et al., 2022; Knorr et al., 2005; Sinsabaugh et al., 2016). Here, we aim to fill this gap by creating a dynamic model embodying multiple N-induced decomposer responses.

#### 1.1 Conceptual model and hypotheses

80

95

In this contribution, we synthesized a model that links soil N availability to the turnover of various C pools, leveraging the stoichiometric control of various decomposer responses. The cornerstone of our model is the choice of relevant variables. We first considered the labile C:N ratio, defined as the ratio of readily-available C and N resource supply to decomposers (Kuzyakov, 2002). We then considered the Threshold Elemental Ratio (TER; Mooshammer et al., 2014a; Sinsabaugh et al., 2013), defined as the ratio of readily-available C and N resources for decomposers at which no C and N limitations and optimal decomposition occur. Conceptually, whenever labile C:N (supply) equals TER (demand), microbial decomposition is at its fastest (Fig. 1; Chen et al., 2014). When labile C:N and TER differ, this constitutes either N-excess (process 1) or N-limitation (process 2), where both conditions reduce decomposition rates, but have opposite response under N addition (Fig. 1).

Labile C:N supply and demand further control microbial turnover since higher N availabilities lowers microbial biomass C:N, favoring fast-turnover copiotrophs and vice versa for oligotrophs (process 3; Fierer et al., 2012; Zhou et al., 2017). Both excess and limitation of N constitute a stoichiometric imbalance, which reduces carbon use efficiency (process 4; Manzoni et al., 2010; Sinsabaugh et al., 2016). Processes 3 and 4 influence microbial biomass (also necromass) production and may jointly explain the commonly observed reduction of microbial biomass under N addition, which may further impact decomposition (Bradford et al., 2017; Lladó et al., 2017; Treseder, 2008; Wu et al., 2023; Xu et al., 2021; process 5, Table 1). Noteworthy is that this stoichiometry-driven model is agnostic about detailed microbial dynamics (explicit changes in allocation to specific

enzymes, enzyme kinetics, community composition and its functional traits, etc.). This is because field measurements are often aggregate outcomes of co-occurring microbial processes that cannot be disentangled (e.g., aggregate element flow, which is in the domain of stoichiometry). We aim to capture the general decadal scale soil C responses to N that could potentially be incorporated into CMIP6 models for predictions in the near-term.

Figure 1. Conceptual model of the hypothesized relationship of microbially-available resource C:N (labile C:N), microbial resource C:N demand (TER), and decomposition rate. Exogenous N addition "pushes" the labile C:N supply curve down. Two general cases emerge: (1) the gap between C:N supply and demand diminishes in soils or substrates that are originally N-poor, e.g., deadwood (brown curve), resulting in decomposition stimulation; or (2) the gap between C:N supply and demand enlarges in soils or substrates that are originally N-rich, e.g., mineral-associated organic matter (blue curve), resulting in decomposition retardation. It is possible for the downward push (N addition) to be large enough to cross from decomposition stimulation to retardation (dashed line).

Based on this conceptual model and our aim, we developed equations capturing these five N-induced decomposer responses, which were implemented in a benchmark base model derived from CENTURY that has no N-induced decomposer response (Parton et al., 2010; Stergiadi et al., 2016). To test the relevance of each hypothesized response, we combined them to create multiple model variants, and tested them against diverse data of experimental N addition responses of multiple C variables, microbial attributes, as well as trends of SOC fractions across environmental nitrogen gradients of Swiss forests. This allowed us to evaluate the relevance of each process simultaneously and propose a coherent mechanism of N effects on the soil C cycle. Our guiding hypothesis is as follows: higher soil N availability (e.g., from exogenous N addition) increases the low C:N pools (e.g., POC, MAOC) via decomposition retardation, but decreases the high C:N detritus (e.g., fresh litter and deadwood) by stimulating its decomposition. This drives changes in the composition of SOC (Fig. 1). We expect this to be captured by some of our new N effect model variants but not by the base model.

## 2 Materials and Methods

## 2.1 General model description

Our base model is derived from CENTURY v4.6 (Parton et al., 1987; Stergiadi et al., 2016), and its parameterization follows ForCent, the forest version of CENTURY (Parton et al., 2010). It is stoichiometry-explicit and simulates C and N pools and fluxes dynamically with a monthly timestep. Litter C input and environmental conditions are model inputs in this study (Fig. 2), except that soil water content is dynamically simulated by a coupled ForClim v4.0.1, a forest model that has been tested extensively across climatic gradients in central Europe (Bugmann, 1996; Huber et al., 2021; Eq. (S12)). The soil C pools of this base model were largely unresponsive to N addition as long as litter input was held constant (see similar findings for the original ForCent model, cf. Savage et al., 2013).

Figure 2. The overall model of this study, its inputs, state variables and processes. Brown solid boxes indicate the monthly updated state variables. All state variables (pools) exist in both the surface and mineral soil layers (0-20 cm), except for SOC3, which only exists in the mineral soil. Green boxes represent inputs to drive the model. Dashed boxes contain soil processes that run at a monthly time step. SOC1, SOC2 and SOC3 denote the carbon of fast-turnover, intermediate-turnover and slow-turnover SOM, respectively.

We briefly describe our base model below, which consists of three basic parts inherited from CENTURY: litter partitioning, decomposition, and boundary mineral N fluxes (Fig. 2). The full model and parameter documentation is available as supplementary materials (Supplement 2).

Litter partitioning transforms the raw litter inputs (leaf, fine root, and twig) into two C pools: recalcitrant "structural" and labile "metabolic" litter based on their lignin:N ratios (Parton et al., 1987; Eq. (S8)). These two litter pools then undergo decomposition. Coarse woody debris has its own state variable pool distinct from structural and metabolic litter.

**Decomposition** follows first-order kinetics with respect to C substrate quantity (Eq. (1 & 2)). Decomposition breaks down structural and metabolic litter, deadwood, and the SOC pools (SOC1, SOC2 and SOC3, approximating microbial biomass C, POC and MAOC, respectively. See Berardi et al. (2024) & Georgiou et al. (2024)), leading to CO<sub>2</sub> release and a transfer of C between pools. In addition to decomposition, a first-order bioturbation further transfers C from surface SOC2 to mineral soil SOC2, following the equation in ForCent (Parton et al., 2010; Eq. (S25)).

$$\frac{dC_x(t)}{dt} = I(t) - gTcflow_x(t)$$
 (1)

$$gTcflow_x = C_x \times kDec_x \times \prod_{i=1}^n f(\varepsilon_i)_x$$
 (2)

where  $C_x$  is a state variable carbon pool x at the current timestep t, I is the C input to  $C_x$  at time t,  $gTcflow_x$  is the total amount of C flowing out of  $C_x$  at time t,  $kDec_x$  is the monthly turnover rate parameter of  $C_x$ , and  $f(\varepsilon_i)_x$  is the ith environmental decomposition modifier (among n modifiers) associated with  $C_x$  at time t. Details can be found in Eq. (S21-25).

The environmental decomposition modifiers reduce the rate of decomposition and bioturbation under suboptimal climatic and edaphic conditions including temperature, relative soil water content, ratio of precipitation to potential evapotranspiration (proxy of anaerobic conditions), pH and sand content (Eq. (S9-20)). The calculation of realized decomposition (Eq. (2)) also leads to net N mineralization or immobilization, determined by the C:N ratio of the decomposing substrate and the computed C:N ratios of new organic matter entering the receiving pool (Eq. (S26-28)).

**Boundary mineral N fluxes** include N deposition, N fixation and soil mineral N loss. Nitrogen deposition is divided into two time periods: 1) *Historical* N deposition calculated by the default equation of CENTURY (Eq. (S38)), and 2) *Contemporary* N deposition based on measured and interpolated data as model input (cf. "*Study sites and model inputs*"). Nitrogen fixation is determined by the central estimate equation in Cleveland et al. (1999) (see also Wieder et al., 2015 and Eq. (S39)). Soil mineral N loss includes leaching and plant uptake, which depends on soil water fluxes and clay content (Eq. (S40-46)).

## 2.2 Base model selection

We refined certain processes and parameters to ensure that our base model reasonably matches observed C stocks, and to reduce biases unrelated to nitrogen. This was necessary as ForCent was originally parameterized and applied on only one site (Parton et al., 2010) and may not capture all the environmental (e.g., soil pH) dependencies of SOC turnover, where pH in particular was shown to be highly influential in Swiss forests (González-Domínguez et al.,2019). To select the base model for subsequent use, we evaluated several model versions: 1. Model with and without a pH retardation effect on decomposition (Eq. (S15)) since it is used in some versions of CENTURY (v4.5 or above, cf. Berardi et al., 2024; Stergiadi et al., 2016); 2. Model with and without a pH retardation effect on bioturbation, since acidity also strongly suppresses faunal bioturbation (Desie et al., 2020; Persson et al., 2007; Taylor et al., 2019), which ForCent did not consider; 3. Alternative turnover rate parameter values for the SOC3 (MAOC) pool (i.e., *kDec5*, see Supplement 2 Table S5), since this parameter value was highly uncertain in ForCent (Parton et al., 2010). We tested the ForCent value (0.0025 y<sup>-1</sup>) against the original CENTURY value (0.0068 y<sup>-1</sup> roughly three times higher; Parton et al., 1987) as preliminary testing showed that ForCent's MAOC turns over very slowly and is unresponsive to environmental perturbations at the decadal scale.

To select the best base model, we compared the simulated outputs averaged over 55-85 years after the onset of contemporary N deposition (roughly corresponding to the timeframe of elevated anthropogenic N emission, cf. *Sect. 2.5* "*Simulation setup*") with observed LFH stocks, SOC stocks (0-20 cm), LFH:SOC and SOC:N (0-20 cm), measured in the 1990s and 2000s in of 16 Swiss ICP Level II long-term forest plots (Thimonier et al., 2005; van der Voort et al., 2016). We evaluated the deviation between simulated and observed data and computed RMSE. The best-performing model was selected as the basis to implement our new N effect models.

## 2.3 Nitrogen-induced decomposer responses

To implement the N effects, we separated the mineral N pools into the surface organic horizon (LFH) and the mineral soil instead of just one homogeneous pool as in the original CENTURY. This allows for calculating separate labile C:N and TER in the surface and mineral soil, and to reconcile simulated mineral N distribution with tracer N studies that tracked the spatiotemporal fate of added N (Li et al., 2019; Templer et al., 2012). Following these modifications, we formulated five equations embodying decomposer adaptive responses to changing N availability. We developed all equations and parameters using data in independent meta-analyses or reviews to maintain model generality, without parameter calibration (see Supplement 2 Sect. 5).

1. **Decomposition retardation of lignin-containing OM and MAOC under N-excess**: We implemented a decomposition retardation effect for lignin-containing pools (structural litter, wood, SOC2) and the protein-rich pool (SOC3) (J. Chen et al., 2018; Chen et al., 2014; Frey et al., 2014; Jian et al., 2016). The retardation effect increases linearly with the extent of N excess (labile C:N < TER, Eq.(3)) until it reaches maximum retardation. The maximum retardation is based on enzyme data reported

in meta-analyses (see Table S5 in Supplement 2). This retardation effect is multiplied with  $gTcflow_x$  same as the other decomposition modifiers in Eq. (2).

$$gNDRF_{x} = \begin{cases} 1 & \text{if } ((gLabileCN_{x} - gTER_{i}) > 0) \\ 1 + \left( (1 - kNDRFmin_{s}) \times \frac{(gLabileCN_{x} - gTER_{i})}{gTER_{i}} \right) & \text{if } ((gLabileCN_{x}) < gTER_{i} \text{ and } gLabileCN_{x} > 0) \\ kNDRFmin_{s} & \text{if } (gLabileCN_{x} = 0) \end{cases}$$

$$(3)$$

where  $gNDRF_x$  is the N-induced decomposition retardation effect for C pool x.  $kNDRFmin_s$  is the maximum reduction (lowest value) for each OM type s (either lignin-containing materials i.e., structural litter, wood, SOC2, or proteinaceous material i.e., SOC3). The extent of retardation is controlled by two variables calculated monthly: 1) labile C:N supply  $(gLabileCN_x)$ , and 2) microbial C:N demand  $(gTER_i)$ . The lower  $gLabileCN_x$  is relative to  $gTER_i$ , the stronger the effect. We formulated this as a linear relationship because the term  $\frac{(gLabileCN_x-gTER_i)}{gTER_i}$  represents the extent of deviation from optimal N demand, which is in the same unit as the extent of decomposition retardation.

Labile C:N surrounding C pool x is calculated by:

210

$$gLabileCN_{x} = \frac{(MetabC_{i} + gTcflow_{x})}{(MetabN_{i} + MinerlN_{i} + gTnflow_{x})}$$
(4)

where the numerator is the sum of all labile C consisting of  $MetabC_i$  – the most labile C pool in layer i (same layer as  $C_x$ ) and  $gTcflow_x$  – the potential decomposable amount of  $C_x$  before N-induced decomposition retardation applies. The denominator is the same but with N, plus an additional mineral N pool in layer i. Metab and MinerlN are assumed to be evenly accessible within the modeled soil layer as our model is in a monthly timestep.  $gLabileCN_x$  is calculated for every instance of  $C_x$  decomposition and hence implicitly accounts for C and N resource heterogeneity in proximity to each OM substrate.

The optimal C:N demand of microbes ( $gTER_i$ ) follows the equation in Sinsabaugh et al. (2013) and Doi et al. (2010):

$$gTER_i = \frac{(SOC1_i/SON1_i)}{kCUEmax}$$
 (5)

where  $SOC1_i/SON1_i$  is the microbial biomass C:N in soil layer i, and kCUEmax is the theoretical thermodynamic maximum CUE (Sinsabaugh et al., 2016). We assumed that the maximum N use efficiency equals to 1 and is thus omitted from the equation (Cui et al., 2023). This optimal TER formulation is necessary as TER is directly compared to labile C:N, instead of bulk C:N.  $gTER_i$  is calculated per layer because there is only one microbial pool per layer, and microbial biomass C:N (hence TER) is inherently less variable than labile C:N due to homeostasis.

2. **Decomposition stimulation of high C:N substrates by alleviating N limitation**: We implemented an explicit mass balance constraint that restricts C decomposition under N limitation (cf. Manzoni & Porporato, 2009). This simultaneously captures N

stimulating the decomposition of N-limited (high C:N) substrates (Allison et al., 2009; Bebber et al., 2011; Micks et al., 2004). Essentially, mineral N shortage restricts the decomposition of high C:N OM into low C:N OM (Eq. (6, 7)):

$$gTcflow\_red_x = \frac{gNdiff \times CN_{donor}}{(CN_{donor}/CNtransfer_{receiver}) - 1}$$
 (6)

where  $gTcflow_red_x$  is the amount of C decomposition to be subtracted from  $gTcflow_x$  (Eq. (2)) based on the shortage of microbially-available N (gNdiff, Eq. (7)), i.e., more available N alleviates the shortage and lowers  $gTcflow_red_x$ .  $CN_{donor}$  is the C:N ratio of the decomposing pool,  $CNtransfer_{receiver}$  is the calculated C:N ratio of new materials entering the receiver pool (Eq. (S26 & S27)). The equation was derived algebraically and conforms to mass balance. gNdiff is calculated as:

$$gNdiff = ((gTcflow_x/CN_{donor}) + MinerlN_i) - (gTcflow_x/CNtransfer_{receiver})$$
 (7)

Note that lignin-containing substrates may experience either decomposition retardation eq. (3) or stimulation eq. (6), as lignin-containing substrates feature a wide range of C:N and hence labile C:N.

3. **Dynamic microbial turnover rate**: We introduced a dynamic turnover rate of the microbial pool, based on the evidence that a lower microbial C:N (under higher N availability) correlates with an increased abundance of fast-turnover copiotrophs, and vice versa for slow-turnover oligotrophs (Fierer et al., 2012; Leff et al., 2015; Rousk and Bååth, 2011). Essentially, the microbial turnover rate  $gDec3_i$  varies linearly between the maximum and minimum microbial C:N (Eq. (8)):

$$gDec3_i = kDec3min_i + \left(kVarat1_1 - (SOC1_i/SON1_i)\right) \times \frac{(kDec3max_i - kDec3min_i)}{(kVarat1_1 - kVarat1_2)}$$
(8)

where  $kDec3min_i$  and  $kDec3max_i$  are the minimum and maximum turnover rates representing oligotrophic and copiotrophic microbial communities respectively, calculated using the ratio of copiotroph and oligotroph turnover rates (see Supplement 2 sect. 5; Rousk and Bååth, 2011).  $kVarat1_1$  and  $kVarat1_2$  are CENTURY's parameters representing the maximum and minimum microbial C:N, respectively.

4. **Dynamic microbial carbon use efficiency**: We introduced dynamic CUE (replacing constant CUE) responsive to the stoichiometric imbalance between labile C:N and TER, approximating the stoichiometric theory in Sinsabaugh et al. (2016). Essentially, CUE decreases in response to an increasing deviation of labile C:N from TER as in  $gNDRF_{\chi}$  (Eq. (3)) until a minimum CUE is reached (Eq. (9)).

$$gCUE_{x} = \begin{cases} kCUEmax & \text{if } (gLabileCN_{x} - gTER_{i} = 0) \\ kCUEmax - \left( (kCUEmax - kCUEmin) \times \frac{(|gLabileCN_{x} - gTER_{i}|)}{gTER_{i}} \right) & \text{if } (0 < |gLabileCN_{x} - gTER_{i}| < gTER_{i}) \\ kCUEmin & \text{if } (|gLabileCN_{x} - gTER_{i}| > gTER_{i}) \end{cases}$$

$$(9)$$

where  $gCUE_x$  is the CUE of the decomposing  $C_x$  (x can be MetabC, StrucC, WoodC, SOC2, SOC3 that decompose to form SOC1), kCUEmax and kCUEmin are parameters representing the maximum and minimum attainable CUE respectively.

5. **Microbial biomass effect on decomposition (MB\_eff)**: We implemented a microbial biomass control on decomposition (Eq. (10)) due to the common co-occurrence of microbial biomass decrease and SOC stock increase under N addition (Lladó et al., 2017; Treseder, 2008; Xu et al., 2021). We assumed that decomposition rate is controlled by "relative microbial biomass" via a Michaelis-Menten type relationship. Relative microbial biomass is calculated as the ratio of realized microbial biomass (SOC1) to the total labile C resources that are potentially convertible to microbial biomass (Eq. (11)), inspired by Guenet et al. (2016), Sulman et al. (2014), and Wutzler & Reichstein (2008). This microbial biomass effect Eq. (10) is multiplied with  $C_x$  like the other decomposition modifiers:

$$gMB\_eff_i = \left(kPrimemax - e^{\left(-\ln\left(\frac{kPrimemax}{kPrimemax-1}\right)\right) \times gRelMBC_i + \ln(kPrimemax)}\right)$$
(10)

where  $gMB_eff_i$  is an asymptotic function controlled by kPrimemax which represents the maximum monthly priming effect (see Supplement 2 for the description of the parameter), and the explanatory variable  $gRelMBC_i$  which represents relative microbial biomass in layer i. This function passes through 1 at  $gRelMBC_i = 1$  and theoretically captures both positive and negative priming when  $gRelMBC_i$  is above or below 1, respectively.  $gRelMBC_i$  is calculated monthly as:

$$gRelMBC_{i} = \frac{SOC1_{i,m}}{\sum_{x,i} C_{x,m-1} \times kDec\_rel_{x,i} \times gCUE_{x,m-1}}$$
(11)

where the denominator is the sum of all C pools (except SOC1) in the last month (m-1) in soil layer i multiplied with their respective relative turnover rate  $kDec\_rel_{x,i}$  and the most recent carbon use efficiency  $gCUE_{x,m-1}$ .  $kDec\_rel_{x,i}$  is defined as the ratio of the decomposition rate parameter of each C pool x relative to the same parameter for metabolic litter (the most labile C) i.e., it extracts the labile portion of each C substrate.

The forms of these equations are presented graphically in Fig. S10 and described in greater detail in Supplement 2 sect. 5. We then combined these five processes (Table 1) into two groups of model variants: those considering decomposition retardation under excess N only (hereafter referred as "Nex<sub>retard</sub> model"), and those considering both retarding (under excess N) and stimulating (under limiting N) effects on decomposition (hereafter referred as "Nex<sub>retard</sub> + Nlim<sub>stim</sub> model"). We then incrementally increased model complexity by including more microbial feedbacks: dynamic turnover, dynamic CUE, and microbial biomass feedback, which resulted in eight model variants to explore the response space (Table 1). This "Lego" incremental modelling approach is similar to the approach by Zhang et al. (2020) and Wutzler & Reichstein (2008).

Table 1. Overview of the five newly implemented N-driven decomposer responses. Different responses are combined to form eight model variants of increasing complexity.

## Nitrogen adaptive responses

| 2. Nlim <sub>stim</sub>                                           | 3. Dynamic microbial                                         | 4. Dynamic CUE                                                                                                                                   | 5. MB_eff                                                                                                                                                                                             |
|-------------------------------------------------------------------|--------------------------------------------------------------|--------------------------------------------------------------------------------------------------------------------------------------------------|-------------------------------------------------------------------------------------------------------------------------------------------------------------------------------------------------------|
| Decomposition<br>stimulation driven by<br>decreasing N-limitation | Microbial turnover rate in response to microbial biomass C:N | Microbial carbon use<br>efficiency in response to<br>N excess or limitation                                                                      | Secondary microbial<br>biomass control on<br>decomposition                                                                                                                                            |
|                                                                   | Decomposition stimulation driven by                          | 2. Nlim <sub>stim</sub> Decomposition  stimulation driven by decreasing N-limitation  turnover  Microbial turnover rate in response to microbial | 2. Nlim <sub>stim</sub> Decomposition  stimulation driven by decreasing N-limitation  decreasing N-limitation  4. Dynamic CUE  Microbial carbon use efficiency in response to  N excess or limitation |

| Model variants combining the adaptive responses in incremental complexity |     |       |         |           |
|---------------------------------------------------------------------------|-----|-------|---------|-----------|
| Model group:                                                              | 1   | 1+3   | 1+3+4   | 1+3+4+5   |
| Nexretard                                                                 | 1   | 1+3   | 1+3+4   | 1+3+4+3   |
| Model group:                                                              | 1+2 | 1+2+3 | 1+2+3+4 | 1+2+3+4+5 |
| $Nex_{retard} + Nlim_{stim}$                                              |     |       |         |           |

## 2.4 Study sites and model inputs

We first tested the base model with data from the Swiss ICP Level II long-term forest plots (n = 16, Thimonier et al., 2005). These plots have comprehensive ancillary data including N deposition and aboveground litterfall, which allowed us to ensure the reliability of our input data during model development. Secondly, to evaluate our N effect models, we used a larger dataset from 54 Swiss forest sites covering wider environmental gradients (e.g., MAT: 1-12 °C; MAP: 587-1847 mm; pH: 3.1-7.6), selected based on maximizing the orthogonality of temperature, moisture, soil, and landform-related variables (González-Domínguez et al., 2019). There are some overlaps between the ICP Level-II plots and the 54 sites in terms of site names, but they belong to separate sampling campaigns differing in time and location.

- For every site, we compiled model input data related to climate, soil, litter, and N deposition to drive the simulations: Monthly temperature and precipitation data were obtained from the 1979-2013 high-resolution monthly climatic data of CHELSA v1.2 (Karger et al., 2017), and extended to long-term climate by stochastic sampling of the monthly climatic data based on their statistical distributions (Huber et al., 2021).
- Contemporary N deposition data are available as measured throughfall N fluxes for most ICP Level II sites (taking the averages from the first sampling year to 2015), and missing data were supplemented with estimates in Thimonier et al. (2005, 2010). For the 54 forest sites without dedicated deposition monitoring, we used interpolated raster data compiled from air pollution inventories of the years 1990, 2000, 2005, and 2010, conducted by the Swiss Federal Office for the Environment (Rihm and Achermann, 2016; Rihm and Künzle, 2023). The reliability of these map data was validated with the measured throughfall data from the aforementioned ICP Level II sites (Pearson r = 0.89, mean error = +3.2 kg N y<sup>-1</sup>. Note the systematic overestimation is because the raster map data represents total N deposition above the canopy, whereas ICP-II measurements are only throughfall N fluxes). These N deposition data are representative of the 1990s and 2000s and comparable to the level in 1940-1970s, with the exception of 1980s when deposition peaked (cf. Gharun et al., 2021; Schöpp et al., 2003).

Soil water holding capacity (*kBS*) down to 1m depth was extracted from the 250 m-resolution map in Baltensweiler et al. (2021). For locations that have no value on the map, we calculated average *kBS* from the eight adjacent cells. Measured soil texture (sand, silt, clay, fine earth density) and pH were obtained from soil inventory data in Switzerland (Walthert et al., 2013).

Litter input (foliage, twig, fine root, exudate, coarse deadwood debris) was generated by the species-explicit ForClim model forced under these climatic and edaphic inputs (Eq. (S3-7)). We took the average litter production of 200 forest patches in the last 100 years of no-management, equilibrium simulations (Bugmann, 1996). Several warm, low-elevation spruce forests resulted from past management practices (Gosheva et al., 2017) were not captured by these natural equilibrium simulations. We therefore re-simulated these sites as pure spruce stands. However, these stands had abnormally low simulated litter production and SOC stocks (likely because reality is not at equilibrium), and hence we increased all litter in these low-elevation spruce stands by a scaling factor of 1.5, according to the measured aboveground litterfall in one of the converted stands (Table S1). We checked the validity of the simulated aboveground litter against plot-level measurements from the ICP Level II sites (Pearson r = 0.79, mean absolute error = 59.6 g C m<sup>-2</sup> y<sup>-1</sup>), as well as the Swiss litter estimates in Gosheva (2017), which showed that our litter inputs were within reasonable ranges (Fig. S1 & S2).

#### 2.5 Simulation setup

We used two sets of simulations to evaluate our N effect model variants. First, we performed "N-addition simulations", where N was added artificially to mimic manipulation experiments. Second, "Contemporary simulations" were run with contemporary N deposition data to reflect real-world conditions (also used in "Base model selection"). In all simulations, we first spinned up the models for 3000 years until all C pools (both in the surface and 20 cm mineral soil layers) were at steady-state. In the Contemporary simulations, we extended the simulation under contemporary N deposition by 85 years, as we took the simulation outputs averaged over 55-85 years (mid-point = 70 years) after the onset of contemporary N deposition for comparing with contemporary observations. In the N-addition simulations, we extended the simulation using contemporary N deposition for 50 years only, and from then on, we added 100 kg N ha<sup>-1</sup> y<sup>-1</sup> on top of the contemporary N deposition as an N treatment to mimic N manipulation experiments. Higher N addition levels were not tested because they are mostly relevant for croplands, not forests (Gundersen et al., 1998).

## 2.6 Model validation and analysis of results

We prepared two types of validation data for the two sets of simulations.

First, to evaluate the outputs of the *N-addition simulations*, we compiled N addition meta-analysis data of the transient responses of various C pools and fluxes (i.e., the percentage difference between N treatment and unmanipulated control). Validation with meta-analysis transient responses was recommended by Wieder, Allison, et al. (2015) and is crucial for making reliable C predictions under global change. The surface organic layer (LFH) responses were extracted from the dataset in Liu & Greaver (2010) and Xu et al. (2021); SOC, POC and MAOC responses were taken from the combined datasets of Tang et

al. (2023) and Wu et al. (2023); soil heterotrophic respiration (Rh) responses were extracted from Liu, Men et al. (2023); microbial biomass (MBC) and microbial biomass C:N (MBC:MBN) responses were combined from Jia et al. (2020) and Zhang et al. (2018). No precise depth information of the microbial samples was available, thus we assumed they were derived from the surface organic horizon and mineral soil equally, for comparison with the simulation results. We further filtered all meta-analysis responses to forest and woodland ecosystems, ≥4 years experimental duration, and a fertilization rate between 40 and 150 kg N ha⁻¹ yr⁻¹. After the filtering, the durations of most N addition experiments were within the range of 5-15 years and hence we averaged simulated responses over this time span for comparison. A summary of the filtered meta-analyses data is provided in Table S2 and Fig. S3.

As the new N equations are hypotheses, we evaluated them by checking whether they produced erroneous response that is far outside the observed range and direction of response, sufficient to be falsified. We are especially interested in matching the changes of different soil C fractions under N addition due to the potential influence of N on OM composition. We also expected that the large environmental gradients in our simulated forest sites (54 Swiss forests) and the meta-analysis samples are comparable qualitatively, given that broad patterns of C pool response are generally consistent across ecosystems and biomes (Ramirez et al., 2012; Wu et al., 2023; Xu et al., 2021). Lastly, we used multiple linear regression to disentangle the contribution of various environmental and experimental factors to the variance of the responses. Predictors with a generalised variance inflation factor (VIF) >10 were removed.

Second, for the *Contemporary simulations*, we evaluated the trends of soil C fractions (hence SOM composition) across soil N gradients. We compiled fraction data consisting of the organic FH horizons, 0-20 cm POC (free + occluded light fraction) and MAOC (clay-associated fine heavy fraction) (González-Domínguez et al., 2019; Griepentrog et al., 2014). FH and POC were further combined into "total light fraction C" to handle sites with no recorded FH layer, and to reduce vertical uncertainties associated with bioturbation and the vertical distribution of root. Specifically, we tested the models' ability to match 1) the proportion of MAOC to total SOC (here defined as the sum of MAOC and light fraction C, without coarse detritus), as well as 2) the proportion of MAOC and light fraction C to total litter input (including coarse detrital inputs). However, unlike N addition experiments isolating the added N as the explanatory factor, these OM proportions arose from a combination of long-term climatic, mineralogic, biotic and management influences. We therefore conducted multiple regression accounting for other site factors to avoid a misattribution of confounding factors (see Table S4). The N gradient was represented by either MAOC:N or light fraction C:N, as they integrate information about the N availability surrounding the respective OM pools over long term.

We first evaluated the simulated MAOC:N and light fraction C:N (e.g., a difference in observed and simulated MAOC:N may imply a mis-calculation of MAOC derived from plant matter vs. microbial products) (Chang et al., 2024). Second, we conducted partial regression of the OM proportions against these fraction C:N ratios to extract their marginal effects, controlling for other site factors. We evaluated the null hypothesis that all model variants and observation are the same

concerning the relationship between OM proportions and OM C:N ratios ( $\alpha = 0.05$ ), handled by a "C:N ratio × Variant" interaction, where "Variant" is a categorical variable that includes both the model ID and observation, and the latter is set as the reference level (see Table S4).

In these analyses, we were mainly interested in the "relationships" between variables (e.g., ratios of C pools, slopes of regressions between X and Y variables; Mahnken et al., 2022; Manzoni & Cotrufo, 2024), instead of matching exact C stock sizes using our no-management, constant litter simulations that must have caused C stocks to somewhat deviate from reality. That is, a close quantitative match of C stocks may not actually be desirable under the uncertainty of simulated site history, an uncertainty ignored by most soil modelling studies.

## 3 Results

#### 3.1 Base model selection

Organic layer (LFH) stocks (Fig. 3a) were generally underestimated by the base model with no pH decomposition retardation effect. Conversely, the models that incorporated it generally overestimated LFH stocks slightly but with an overall much lower RMSE and slope closer to 1. Including the pH effect on bioturbation similarly increased LFH stocks but at a smaller magnitude.

All models generally overestimated mineral topsoil SOC stocks (Fig. 3b), and the inclusion of the pH decomposition retardation effect increased this overestimation. Models adopting the high SOC3 turnover parameter value reduced the RMSE substantially.

All models generally underestimated the LFH:SOC ratios (Fig. 3c), especially those with the low SOC3 turnover parameter. Including the pH effect on bioturbation consistently reduced the RMSE of LFH:SOC. For SOC:N ratios (Fig. 3d, probably the most relevant performance metric for our stoichiometric model), they were largely underestimated by models with no pH effect on decomposition, but their performance was consistently improved when including both the pH effects and the high SOC3 turnover parameter value (mean SOC:N = 15.4, RMSE<sub>SOCN</sub> = 4.15).

In sum, the best base model with low RMSE in general included the pH effects on decomposition and bioturbation, and the high SOC3 turnover parameter. Hence, this combination of processes and parameters was included in the base model for the subsequent N effect simulations. The resultant distributions of simulated SOC1, SOC2, SOC3 stocks and their turnover times of this base model were reported in Fig S4.

Figure 3. Simulated vs. observed (a) LFH C stock, (b) SOC stock (0-20 cm), (c) LFH:SOC ratio, (d) SOC:N ratio (0-20 cm) at the Swiss ICP Level II sites (n = 16) among base model variants: with or without a pH effect on decomposition, with or without a pH effect on bioturbation, and with the ForCent low SOC3 turnover (0.0025 y<sup>-1</sup>) or the CENTURY high SOC3 turnover parameter (0.0068y<sup>-1</sup>). Ellipses show the 2D range of 95% confidence ellipses (Fox and Weisberg, 2011).

### 3.2 Nitrogen addition simulation: soil C-cycle responses

The meta-analysis data showed that on average, LFH increased (+27.6%) more than mineral soil SOC (+9.9%), and POC increased (+19.7%) more than MAOC (+7.8%) under N addition across global forests (Fig. 4abcd). These patterns were similar in temperate forests, and were best reproduced by the Nex<sub>retard</sub> + Nlim<sub>stim</sub> model variants (i.e., considering decomposer response to both N-excess and N-limitation), but neither by the base model (0% response) nor the simple Nex<sub>retard</sub> models without

decomposition stimulation, dynamic CUE and microbial biomass feedback (MB\_eff). The simple Nex<sub>retard</sub> models produced negative LFH response that were well below the meta-analysis lower quartiles (Fig. 4a). Conversely, the Nex<sub>retard</sub> + Nlim<sub>stim</sub> model variants yielded responses that were generally between the meta-analysis means and lower quartiles, with the exception of LFH response in temperate forests (Fig. 4abcd). It must be noted however that LFH response was strongly reduced by overestimated initial LFH stocks in the Nex<sub>retard</sub> + Nlim<sub>stim</sub> models (Table 2 & Fig. 3). In all model variants, MAOC was largely unresponsive to N addition (Fig. 4d). However, the MAOC response of the Nex<sub>retard</sub> + Nlim<sub>stim</sub> variants continued to increase on a longer time scale beyond the 5-15 year period (Fig. S5), nearing the meta-analysis response of temperate forests. The time courses of the other responses were presented in Fig. S6 for reference.

For the microbial attributes (Fig. 4ef), the base model had no microbial biomass response but a strongly negative microbial biomass C:N response, which the simple Nex<sub>retard</sub> models also predicted. The other more complex models had milder microbial C:N responses which were close to the meta-analysis mean and lower quartile. Incorporating dynamic microbial turnover consistently reduced microbial biomass (Fig. 4e). The microbial biomass responses were biased towards the lower and upper quartiles respectively for the Nex<sub>retard</sub> only and Nex<sub>retard</sub> + Nlim<sub>stim</sub> models. However, it must be noted that the simulated positive microbial biomass responses were mainly from the surface horizon, whereas experimental measurements likely included more mineral soil samples.

For the deadwood and litter responses which we had insufficient observation data (Fig. 4gh), the Nex<sub>retard</sub> + Nlim<sub>stim</sub> models featured large negative responses (decomposition stimulation). Including more microbial feedbacks (dynamic CUE and MB\_eff) attenuated these negative responses and further drove them to positive ranges in the simple Nex<sub>retard</sub> models. Lastly, total dead organic C stock responded in a similar direction to the deadwood response but opposite direction to the Rh responses (Fig. 4j). All models reproduced Rh within the meta-analysis interquartile range but the Nex<sub>retard</sub> + Nlim<sub>stim</sub> models with microbial feedbacks were the closest to the mean, as well as capturing a larger variance (Fig. 4i).

Figure 4. Global forest observed responses and simulated responses of 54 Swiss forest sites under N addition: % difference in N treatment vs. control of (a) LFH C stock, (b) mineral soil POC stock, (c) mineral soil SOC stock, (d) mineral soil MAOC stock, (e) MBC (surface + mineral soil), (f) MBC:MBN (surface + mineral soil), (g) surface fresh litter C stock, (h) deadwood C stock, (i) annual heterotrophic respiration (surface + mineral soil), and (j) total dead organic C stocks in the base model and eight model variants averaged over the year 5-15 after the start of N addition. The solid orange vertical lines are the observed global meta-analysis means, and the dot-dash lines are the lower and upper quartile values. The green shaded region consists of a subset of temperate forests only (by IGBP-DIS classification (Woodward et al., 2004). N is the global forest sample size and n is the temperate forest.

We further analyzed the site factors contributing to the variability of the N response in Fig. 4. We found that the Nex<sub>retard</sub> only and the Nex<sub>retard</sub> + Nlim<sub>stim</sub> model variants formed two distinct groups, each influenced differently by the site factors (Table 2).

In the Nex<sub>retard</sub> models, most environmental factors were not significant, reflecting the small inter-site variance in their predictions. An exception is ambient N deposition, which had significant positive effects on various C pools but a negative effect on deadwood. In contrast, in the Nex<sub>retard</sub> + Nlim<sub>stim</sub> models, N deposition had significant negative effects on the response of SOC, POC and R<sub>h</sub>, while positively affecting deadwood. Climatic effects were notable as MAT (warmer) and MAP (wetter) showed significant positive effects on the simulated response of LFH, SOC, POC, MAOC, MBC and Rh, but a negative effect

on deadwood. This aligns with the higher observed responses of LFH, SOC, MAOC in warmer climate (Table S3). The observed LFH response correlates negatively with MAP in temperate forests, which did not align with any model. Furthermore, a clear opposite direction of the environmental effects emerged between deadwood and other C pools in the Nex<sub>retard</sub> + Nlim<sub>stim</sub> models. Lastly, the percentage responses of LFH, SOC and POC were reduced by larger (i.e., overestimated) initial C stocks (cf. Fig. 3), also only in the Nex<sub>retard</sub> + Nlim<sub>stim</sub> models.

430

Table 2. Multiple-regression of the site factors that control simulated % response of selected C-cycle variables under N addition. Standardized coefficients are scaled by their standard deviation.

| LFH response                  | Nex <sub>retard</sub> models |           | Nex <sub>retard</sub> + Nlim <sub>stim</sub> models |           |
|-------------------------------|------------------------------|-----------|-----------------------------------------------------|-----------|
| Predictors                    | Std. Coefficient             | P-value   | Std. Coefficient                                    | P-value   |
| Intercept                     | NA                           | 0.04*     | NA                                                  | 0.5       |
| MAT                           | -0.12                        | 0.19      | 0.17                                                | 0.03*     |
| MAP                           | -0.28                        | <0.001*** | 0.18                                                | <0.001*** |
| sand                          | -0.02                        | 0.76      | 0                                                   | 0.98      |
| Ambient Ndep†                 | 0.14                         | 0.12      | 0.14                                                | 0.06.     |
| Broadleaf: conifer litterfall | 0.08                         | 0.15      | -0.13                                               | 0.01**    |
| Initial LFH                   | -0.18                        | 0.53      | -1.18                                               | <0.001*** |
| Initial deadwood              | 0.24                         | 0.38      | 1.22                                                | <0.001*** |
| SOC response                  | Nex <sub>retard</sub> models |           | Nex <sub>retard</sub> + Nlim <sub>stim</sub> models |           |
| Predictors                    | Std. Coefficient             | P-value   | Std. Coefficient                                    | P-value   |
| Intercept                     | NA                           | 0.39      | NA                                                  | 0.67      |
| MAT                           | -0.04                        | 0.62      | 0.49                                                | <0.001*** |
| MAP                           | 0.05                         | 0.41      | 0.45                                                | <0.001*** |
| sand                          | 0.05                         | 0.58      | -0.01                                               | 0.88      |
| Ambient Ndep†                 | 0.17                         | 0.03*     | -0.27                                               | <0.01***  |
| Broadleaf: conifer litterfall | -0.14                        | 0.01**    | -0.11                                               | 0.05*     |
| Initial SOC                   | -0.01                        | 0.93      | -0.61                                               | <0.001*** |
| Initial deadwood              | 0.01                         | 0.96      | 0.26                                                | 0.05.     |
| POC response                  | Nex <sub>retard</sub> models |           | $Nex_{retard} + Nlim_{stim}$ models                 |           |
| Predictors                    | Std. Coefficient             | P-value   | Std. Coefficient                                    | P-value   |
| Intercept                     | NA                           | <0.01***  | NA                                                  | 0.71      |
| MAT                           | -0.03                        | 0.72      | 0.46                                                | <0.001*** |
| MAP                           | 0.05                         | 0.38      | 0.43                                                | <0.001*** |
| sand                          | 0.24                         | 0.01**    | -0.02                                               | 0.72      |
| Ambient Ndep†                 | 0.24                         | 0.01**    | -0.27                                               | <0.01***  |
| Broadleaf: conifer litterfall | -0.16                        | 0.01*     | -0.16                                               | 0.01**    |
| Initial POC                   | 0.02                         | 0.96      | -0.64                                               | <0.001*** |
| Initial deadwood              | 0.09                         | 0.81      | 0.22                                                | 0.11      |
| MAOC response                 | Nex <sub>retard</sub> models |           | Nex <sub>retard</sub> + Nlim <sub>stim</sub> models |           |

| Predictors                    | Std. Coefficient             | P-value   | Std. Coefficient                                    | P-value   |  |
|-------------------------------|------------------------------|-----------|-----------------------------------------------------|-----------|--|
| Intercept                     | NA                           | <0.001*** | NA                                                  | 0.6       |  |
| MAT                           | -0.07                        | 0.33      | 0.36                                                | <0.001*** |  |
| MAP                           | -0.08                        | 0.24      | 0.09                                                | 0.21      |  |
| sand                          | -0.23                        | <0.001*** | 0.01                                                | 0.95      |  |
| Ambient Ndep†                 | 0.09                         | 0.24      | -0.05                                               | 0.54      |  |
| Broadleaf: conifer litterfall | -0.01                        | 0.86      | -0.06                                               | 0.29      |  |
| Initial MAOC                  | 0.01                         | 0.83      | 0.13                                                | 0.09.     |  |
| Initial deadwood              | -0.17                        | <0.001*** | -0.4                                                | <0.001*** |  |
| MBC response                  | Nex <sub>retard</sub> models |           | Nex <sub>retard</sub> + Nlim <sub>stim</sub> models |           |  |
| Predictors                    | Std. Coefficient             | P-value   | Std. Coefficient                                    | P-value   |  |
| Intercept                     | NA                           | 0.03*     | NA                                                  | 0.04*     |  |
| MAT                           | -0.03                        | 0.7       | 0.32                                                | <0.001*** |  |
| MAP                           | -0.1                         | 0.03*     | 0.3                                                 | <0.001*** |  |
| sand                          | 0.08                         | 0.12      | -0.02                                               | 0.74      |  |
| Ambient Ndep†                 | 0.19                         | 0.01**    | -0.13                                               | 0.07.     |  |
| Broadleaf: conifer litterfall | 0.03                         | 0.46      | -0.09                                               | 0.07.     |  |
| Initial MBC                   | 0.25                         | 0.04*     | -0.62                                               | <0.001*** |  |
| Initial deadwood              | 0.09                         | 0.44      | 0.65                                                | <0.001*** |  |
| Deadwood response             | Nex <sub>retard</sub> models |           | $Nex_{retard} + Nlim_{stim}$ models                 |           |  |
| Predictors                    | Std. Coefficient             | P-value   | Std. Coefficient                                    | P-value   |  |
| Intercept                     | NA                           | <0.001*** | NA                                                  | 0.78      |  |
| MAT                           | 0.05                         | 0.46      | -0.61                                               | <0.001*** |  |
| MAP                           | -0.06                        | 0.2       | -0.34                                               | <0.001*** |  |
| sand                          | -0.13                        | <0.01***  | 0.06                                                | 0.27      |  |
| Ambient Ndep†                 | -0.14                        | 0.04*     | 0.37                                                | <0.001*** |  |
| Broadleaf: conifer litterfall | 0.03                         | 0.51      | 0.09                                                | 0.14      |  |
| Initial deadwoodC             | -0.22                        | <0.001*** | 0.27                                                | <0.001*** |  |
| Rh response                   | Nex <sub>retard</sub> models |           | Nex <sub>retard</sub> + Nlim <sub>stim</sub> models |           |  |
| Predictors                    | Std. Coefficient             | P-value   | Std. Coefficient                                    | P-value   |  |
| Intercept                     | NA                           | 0.99      | NA                                                  | 0.87      |  |
| MAT                           | -0.07                        | 0.35      | 0.17                                                | 0.05.     |  |
| MAP                           | 0.07                         | 0.19      | 0.19                                                | <0.01***  |  |
| sand                          | -0.02                        | 0.69      | 0.11                                                | 0.05*     |  |
| Ambient Ndep†                 | -0.04                        | 0.57      | -0.21                                               | 0.01*     |  |
| Broadleaf: conifer litterfall | 0.05                         | 0.32      | 0.03                                                | 0.61      |  |
| Initial deadwoodC             | 0.32                         | <0.001*** | 0.53                                                | <0.001*** |  |

<sup>\*</sup> For P-values, one asterisk indicates p < 0.05, two asterisks indicate p < 0.01, three asterisks indicate p < 0.001.

<sup>435 †</sup> Ambient Ndep is the contemporary N deposition level without artificial N addition.

## 3.3 Contemporary simulation: Organic matter proportions across Swiss forests

First, we evaluated MAOC:N and light fraction C:N as we used them as proxies that reflect the N-status of soils. The models captured the general trend of MAOC:N but there was a cluster of underestimations associated with broadleaved forests (R<sup>2</sup> = 0.33, P < 0.001, Fig. 5a). However, it matched the observed correlations with environmental factors fairly (Fig. 5c). This was not the case for light fraction C:N ratios where models underestimated them substantially and did not vary properly with environmental factors (Fig. 5bd). Lower MAOC:N (N-rich) was mainly associated with warmer, high pH, high CEC and clayey sites (Fig. 5c). The observed light fraction C:N had a negative correlation with MAT, MAP and N deposition, which could not be captured by the simulation (Fig. 5bd).

Second, we evaluated the partial regression relationships between OM proportions and these C:N ratios (holding other predictors at their means). The observed proportion of MAOC to total SOC (coarse detritus not included) showed a significant negative relationship with MAOC:N (i.e., lower MAOC:N associating with more MAOC; Fig. 6a). The slope of this observed relationship was matched by the Nex<sub>retard</sub> + Nlim<sub>stim</sub> models with dynamic CUE and microbial biomass feedback enabled (Table S4, p > 0.05). The observed ratios of MAOC: annual litter showed a positive but non-significant relationship with MAOC:N (Fig. 6b). This was again echoed only by the microbial model variants considering dynamic CUE and microbial biomass feedback (Table S4, p > 0.05), and the other models deviated substantially from the observations. Lastly, the light fraction C: annual litter ratios had non-significant relationships with light fraction C:N in both observations and simulations.

Figure 5. A comparison of simulated and observed (a) MAOC:N, (b) Light Fraction C:N ratios, and (c & d) their respective correlations to environmental factors (\*\* are correlations with P-value 

Figure 6. Partial regression plots of organic matter proportions against various C:N ratios (holding other predictors at mean levels): (a) MAOC: total SOC vs. MAOC:N, (b) MAOC: annual litter input vs. MAOC:N (c) Light Fraction C: annual litter input vs. light fraction C:N, of the 54 Swiss forest sites. Total SOC is defined as MAOC plus light fraction C (excluding coarse woody detritus, which was not measured), and annual litter inputs include all litter both woody and non-woody. In the regressions, the model variants together with observations constitute a categorical variable that interacts with the C:N ratios to detect differences between models and observations. The full multiple regression equations are documented in Table S3 and the adjusted R<sup>2</sup> and P-values in the plots correspond to these full regressions.

## 4. Discussion

To our knowledge, this is the first study that elucidates the diverse effects of N on multiple soil C pools, microbial attributes, and Rh across a large number of sites, with a hypothesis-driven model experiment. Previous studies typically adopted a simple parameter adjustment to represent a one-time, static decomposition retardation effect (Chen et al., 2019; Eastman et al., 2024; Tonitto et al., 2014; Whittinghill et al., 2012), or derived theoretical eco-enzymatic models responsive to nutrients, which contain many detailed processes and parameters that are difficult to estimate or constrain (Moorhead & Sinsabaugh, 2006; Schimel & Weintraub, 2003; Sinsabaugh & Shah, 2012; Wutzler et al., 2017). Our models represent a compromise between the two with the use of simple stoichiometric principles, taking into account the data available for use to develop models.

## 4.1 Base model performance

The base model over-estimated topsoil SOC stocks, which have various potential causes. First, simulated C turnover rates may be too low, as the overestimation was greatly alleviated by an increase in SOC3 (MAOC) turnover. The simulated turnover times of MAOC ranged from 550 - 1500 y, which may be too high as other studies have reported turnover times as low as

decades (Guo et al., 2022; Kleber et al., 2015; Lavallee et al., 2020). Increasing MAOC turnover also improved SOC:N ratios (which largely correlate with POC:MAOC ratios), implying this adjustment was in the right direction. Besides, the overestimated SOC stocks may arise from the assumption of no-management, as most Swiss forests have undergone past management that likely reduced C stocks to some extent (e.g., logging, deadwood removal, litter raking; Gimmi et al., 2013; Gosheva et al., 2017; Risch et al., 2008; Wäldchen et al., 2013). Furthermore, SOC stocks generally have longer turnover (hence recovery) times than LFH upon the cessation of management or disturbance (Hiltbrunner et al., 2013; Schulze et al., 2009), which may simultaneously explain why observed LFH:SOC tended to be higher than simulated LFH:SOC.

Modelling the effects of pH on soil C cycling is important but remains under-studied (Leifeld et al., 2008; Sinsabaugh et al., 2008; Wieder et al., 2013, 2015b). The contradictory results of the pH retardation effect on decomposition (improving LFH but worsening SOC stocks) imply that the pH effect in CENTURY is likely not formulated properly. CENTURY assumes that all C pools experience the same pH (including coarse wood logs). In reality, the F and H horizons are usually the most acidic (Takahashi, 1997), more acidic than fresh detritus (Burgess-Conforti et al., 2019; Khanina et al., 2023) and mineral soils with increasing depth (Iwashima et al., 2012; Solly et al., 2020). Hence, the pH retardation effect may be weaker outside the FH horizons and a more realistic, heterogeneous pH effect may be necessary. Moreover, pH has complex interactions with soil minerals and polyvalent ions to stabilize OM (Rowley et al., 2018; Solly et al., 2020; Ye et al., 2018), which is not considered in most models (Sokol et al., 2022).

In summary, the selected base model improved several variables when incorporating the pH effect on decomposition and bioturbation, and enhanced SOC3 turnover, but it overestimated topsoil SOC primarily under acidic conditions (irrespective of nitrogen effects). Nonetheless, we emphasize that our focus is the response of various C pools under changing N availability, and removing acidic sites did not affect the general response pattern (Fig. S7). In fact, overestimated C stocks contributed to smaller percent responses especially for Nex<sub>retard</sub> + Nlim<sub>stim</sub> models (Table2), so their pattern of response improved when the overestimated sites were removed.

#### 4.2 Exogenous nitrogen effects on soil organic matter

Under N addition, model variants considering both decomposition retardation and stimulation (i.e., response to both N-excess and N-limitation, i.e., Nex<sub>retard</sub> + Nlim<sub>stim</sub>) reproduced the closest match to the patterns of a larger increase of LFH than SOC, and POC than MAOC under N addition (cf. Liu & Greaver, 2010; Tang et al., 2023; Wu et al., 2023; Xu et al., 2021). These patterns aligned partially with our guiding hypothesis (Fig. 1). Mechanistically, they can be explained as follows: POC and FH (i.e. light Fraction), not MAOC, receive C directly from detrital (e.g. deadwood) decomposition, which is stimulated by N addition (Allison et al., 2009; Bebber et al., 2011; Błońska et al., 2019; Kuyper et al., 2024; Lagomarsino et al., 2021). As a corollary, MAOC experiences smaller changes and relies more on decomposition retardation. In addition, the surface receives direct N addition and contain more detritus than the mineral soil (He et al., 2018; Kurz et al., 1996). Consequently, the surface tends to experience stronger N effects, explaining a larger increase in the surface organic layer than mineral soil SOC.

In summary, N addition (range ≈ 40-150 kg N ha<sup>-1</sup> y<sup>-1</sup>) accelerates the decomposition of N-limited detritus, where the processed C becomes plant-derived FH and POC pools (i.e., light fraction C). Furthermore, N retards the decomposition of all processed organic matter with lower C:N ratios i.e., both light fraction and MAOC (Fig. 7). Although we did not have sufficient data to evaluate the response of detritus decomposition, this mechanism has been corroborated by many studies. Berg & Matzner (1997) and Knorr et al. (2005) found that the decomposition of fresh, N-limited litter was stimulated by N addition, contrary to the retardation in older, processed litter humus more enriched in lignin and nitrogen. Chen et al. (2015) found that N addition retarded litter decomposition in N-rich forests but not in N-limited forests in China. Bonanomi et al. (2017) found that high C:N substrates decomposed faster when inoculated to N-rich soils, but not for initially low C:N litter. Aside from leaf litter (the most commonly experimented litter), Allison et al. (2009) clearly demonstrated that wood decomposition was also stimulated by N addition at low dose (< 100 kg N ha<sup>-1</sup> y<sup>-1</sup>).

Many past studies attributed SOC accumulation under N addition to decomposition retardation alone through reduced microbial biomass (Lladó et al., 2017; Treseder, 2008; Wu et al., 2023; Xu et al., 2021). If lignin-containing materials (including fresh detritus) only experience decomposition retardation as hypothesized by other studies, all the downstream C accumulation would be difficult to explain, as light fraction (FH and POC) receives C input from detrital decomposition (cf. Fig. 4a & 4g), and MAOC receives C from light fraction decomposition and DOC from litter decomposition (Angst et al., 2021; Bramble et al., 2024; Chang et al., 2024; Forstner et al., 2019; Whalen et al., 2022). Moreover, Rh was not reduced significantly in either simulations or meta-analysis, further indicating that retardation may not be the sole mechanism. Considering the N-induced decomposition stimulation of N-limited detritus can resolve the inconsistency, a process absent in most soil models (Manzoni and Porporato, 2009).

Decomposer N-limitation responses (i.e., either Nlim<sub>stim</sub> or N-responsive CUE) is also the basis of microbial biomass C:N homeostasis (cf. Manzoni & Porporato, 2009). In the absence of Nlim<sub>stim</sub> or N-responsive CUE, the simple Nex<sub>retard</sub> models produced erroneous response under N addition (e.g., strongly negative LFH and microbial C:N response). The latter indicates a failure of microbial biomass C:N (hence TER) homeostasis, which paradoxically led to a larger decrease of TER than labile C:N under N addition (decreasing N excess, data not shown). That is, microbial C:N homeostasis is needed to properly constrain microbial N-limitation and excess. Hence, incorporating N-limitation microbial response in soil models may be necessary, but it remains unclear whether current models have improved in this regard since the seminal review conducted by Manzoni & Porporato (2009).

Figure 7. A synoptic overview of the main decomposition effects of boundary N inputs on various organic matter pools. The illustrated effect sizes depict the general trends concluded from this study, but it must be noted that the effect varies for individual soils and organic matter that has a wide range of C:N ratios.

Faster microbial turnover drove the microbial biomass reduction under N addition, with minimal impacts on other C pools. A shift towards copiotrophic communities is indeed commonly observed under N addition (Fierer et al., 2012; Gravuer and Eskelinen, 2017; Leff et al., 2015; Zechmeister-Boltenstern et al., 2011; Zhou et al., 2017). Enabling dynamic CUE (mainly CUE reduction) also tended to reduce microbial biomass, but this reduced the size of other C pools (light fraction and MAOC) that receive necromass (cf. Wang et al., 2021). In turn, enabling microbial biomass control (MB\_eff) somewhat counteracted this reduction: as CUE decreases, biomass (and necromass) decreases, and a lower microbial biomass retards decomposition. This pathway of feedback to C cycle corresponds to the balance of "microbial priming vs. necromass entombment effect" in Liang et al. (2017). Altogether, enabling these microbial feedbacks mainly improved the responses of microbial attributes and Rh, but did not alter the main patterns of soil C pools driven by Nlim<sub>stim</sub>.

Lastly, individual soil C responses varied with environmental factors despite the general pattern of response (cf. Table 2). Notably, we found that cold soils were generally more prone to decomposition retardation. Indeed, a number of studies showed that colder, particularly boreal sites experienced more Rh reduction than warm sites (Chen et al., 2023; Guo et al., 2023; Marshall et al., 2021; Zhong et al., 2016). The likely explanation is that colder sites have smaller basal N fluxes, which translate to a larger N excess (i.e., retardation) when a fixed amount of N was added in experiments. In parallel, N addition favors

copiotrophs and suppresses oligotrophic, ectomycorrhizal communities more common in cold forests, resulting in a stronger decomposition retardation of recalcitrant substrates (Kuyper et al., 2024; Lilleskov et al., 2019; Zhou et al., 2017). Although microbial functional groups are not explicitly represented in our models, oligotrophic decomposers in cold soils correlate with a higher biomass C:N and TER, which led to a higher chance of N-excess in our model. In contrast, in warm sites, the N-induced responses of soil C were driven primarily by the decomposition stimulation of detritus. Decomposers have high potential activities in warm favorable conditions but are limited by N in detritus (Allison et al., 2009; Cai et al., 2024; Maslov and Maslova, 2021). As a result, N addition releases this limitation and promotes decomposition.

## 4.3 Contemporary nitrogen effect on soil organic matter

Despite the poorly simulated light fraction C:N, we found strong negative relationships between MAOC:N and MAOC:total SOC in both observations and models (here, total SOC = MAOC + light fraction C). We interpret this as N-rich (low MAOC:N) soils favoring the formation and/or protection of MAOC in the long term. First, N-rich, labile litter contains more water-soluble compounds (instead of structural compounds) that may readily form MAOC (instead of the light fraction C) (Cotrufo et al., 2013, 2015; Sokol et al., 2022). This is accounted for in the base model, hence all models inherited this negative relationship. Second, the remaining difference between models is driven by how N influenced MAOC and light fraction C accumulation, where 1) the Nex<sub>retard</sub> + Nlim<sub>stim</sub> models favored light fraction, but 2) the Nex<sub>retard</sub> models favored MAOC at low MAOC:N. Our results support that light fraction C may accumulate more readily than MAOC under the N conditions in Swiss forest soils, similar to the findings regarding N addition.

However, this negative trend disappears when considering the relationship between MAOC:N and MAOC:total litter C input

an indicator of the overall conversion efficiency of detrital C to MAOC and its persistence. The base model and simple
Nex<sub>retard</sub> models clearly overestimated MAOC conversion/protection at low MAOC:N and falsely retained the negative trend.
This was substantially alleviated in the Nex<sub>retard</sub> + Nlim<sub>stim</sub> models with microbial feedback enabled, again highlighting their importance and that retardation effect cannot be solely considered. Essentially, enabling N limitation and microbial feedback
limits OM processing, thereby reducing the conversion to MAOC. Nonetheless, no model can capture MAOC stocks in high
MAOC:N soils (coinciding cold, coniferous, acidic soils). Several studies found that leaching of dissolved organic C from
coniferous forest floors throughout the year (not just in autumn) was conducive to MAOC formation (Andivia et al., 2016;
Bramble et al., 2024; Córdova et al., 2018). Our models do not contain dissolved C dynamics, which may imply a need for
further model improvement as fresh litter decomposition and hence leaching from the litter layer is likely enhanced by N
(Hagedorn et al., 2012b).

Overall, these findings under contemporary N conditions tend to converge with the N addition findings prior (e.g., larger light fraction C accumulation, which is captured by the Nex<sub>retard</sub> + Nlim<sub>stim</sub> models). However, not surprisingly, the differences between model variants were smaller, as N deposition is typically smaller (6-38 kgN ha<sup>-1</sup> y<sup>-1</sup>) than the amount of N added in experiments (40-150 kgN ha<sup>-1</sup> y<sup>-1</sup>). Multiple meta-analyses have found dose-dependent responses of both soil and plant

variables, where low N addition (<50 kgN ha<sup>-1</sup> y<sup>-1</sup>) tended to produce small effects not different from controls (Chen et al., 2015; Jian et al., 2016; Liu et al., 2024; Tang et al., 2023; Xu et al., 2021). Nonetheless, without accurate and complete boundary N input data (N deposition, N fixation, and even pedogenic N sources, cf. Houlton et al., 2018), it remains difficult to evaluate the true impact of contemporary N gradients on soil organic matter.

#### 4.4 Limitations

- We acknowledge that the simulated N-addition responses are only qualitatively comparable to the meta-analysis responses, as their environmental gradient coverage differs (cf. Table S2). However, we found a strong consistency in the general response pattern regardless of the forest biome classification (cf. Fig. 4. See also Ramirez et al., 2012). Nonetheless, we urge more experiments to be conducted outside the temperate zone where data is still limited. Moreover, deadwood data remains scarce but highly valuable, as we showed that deadwood response may influence the responses of FH and POC pools and the overall belowground C balance strongly. More investigations measuring detritus and in ecosystems covering a wider woodiness (hence C:N) gradient (e.g., close-canopy forest -> savannah -> grassland) would be highly valuable. Overall, although Nex<sub>retard</sub> + Nlim<sub>stim</sub> models can match the responses reasonably, the meta-analyses responses generally featured a larger variation than simulation, due to its larger environmental coverage, along with the use of different N forms, application rates and frequencies (Du et al., 2014; Lu et al., 2021; Xu et al., 2021), as well as possible plant feedback not considered here.
- Causal attribution is another challenge. Feedback to plant litter production and soil acidification are two other possibilities that could influence the N effects on soil C. First, N addition generally increases aboveground litterfall, which would increase and potentially improve the LFH response (Chen et al., 2015; Janssens et al., 2010). However, litterfall alone may not explain the large LFH increase. Several studies (Bowden et al., 2019; Frey et al., 2014; Lovett et al., 2013) reported no significant change in aboveground litterfall yet observed substantial LFH increases. For example, Bowden et al. (2019) observed an 80% increase 610 in the organic horizon of a mature hardwood forest after 17 years of N addition, which could be attributed neither to litterfall (no significant change) nor to suppressed soil respiration (only -5.7%). Schulte-Uebbing and de Vries (2018) reported that increased aboveground production was mainly associated with young stands where experiments are more common, and hence more studies should be conducted in mature forests. Root litter response ranged from a decrease to an increase, leading to uncertain effects on both FH and SOC stocks (Chen et al., 2015; Feng et al., 2023; Franklin et al., 2003; Frey et al., 2014; Peng 615 et al., 2017; Yue et al., 2021). Again, this highlights the need for studies on how N influences root litter production and their modulating factors. Once these are better quantified, incorporating the litter response into new models is highly valuable (Xia et al., 2018), as the Nlim<sub>stim</sub> effect by definition interacts with the amount of litter inputs, where more litter may amplify the role of N-limitation.
- Second, acidification suppresses decomposition (Averill and Waring, 2018; Tian and Niu, 2015; Wu et al., 2023; Ye et al., 2018). Our minimum Nex<sub>retard</sub> parameter (i.e., maximum retardation effect) does not distinguish the cause of decomposition decline since N addition rate, N-surplus and pH decrease are generally collinear (Tian and Niu, 2015; Zhou et al., 2017), and

hence our Nex<sub>retard</sub> effect may implicitly include an acidification effect already. Nonetheless, acidification alone cannot explain any process other than decomposition retardation (e.g., stimulation). Even for retardation effect, Frey et al. (2025) showed that it was mainly driven by increasing N availability in a field experiment designed to disentangle N and acidification effects. Besides, studies had shown that nitrogen and acidification led to different microbial community responses. Nitrogen addition predominantly promoted copiotrophic taxa, replacing oligotrophic bacteria (e.g., Acidobacteria) and fungi selected under acidic conditions that are better adapted at decomposing recalcitrant substrates (Bardhan et al., 2012; Choma et al., 2020; Fierer et al., 2012; Lauber et al., 2009; Zhou et al., 2017). Ramirez et al. (2010) further found that even alkaline N compounds retarded decomposition. Therefore, N likely has a stronger, direct selection effect on microbes and their enzymes, and N-induced acidification effects are secondary unless for extreme acidification (e.g., in agricultural lands), a conclusion also supported by the meta-analysis of Zhou et al. (2017).

Another uncertainty is the lack of MAOC response in our N addition simulations (Fig. 4D). However, MAOC did respond on a longer time scale in the Nex<sub>retard</sub> + Nlim<sub>stim</sub> models (cf. Fig. S5), matching the mean response in temperate forests. Nonetheless, the large variation of meta-analysis MAOC response could not be captured. A possible reason is that experimentally fractionated MAOC contains sub-fractions of labile, fast-turnover compounds not represented by the modelled "homogeneously-stable" MAOC (Brunmayr et al., 2024; Guo et al., 2022; Poeplau et al., 2018; Schrumpf and Kaiser, 2015; Sokol et al., 2022). For Swiss forests, Moreno-Duborgel et al. (2025) subdivided MAOC further into a peroxide-oxidizable and residual (resistant) fractions and showed distinct turnover times of 32–1440 and 130–4597 y, respectively. The turnover times of MAOC also vary considerably from decades to millennia in the literature (Guo et al., 2022; Kleber et al., 2015; Lavallee et al., 2020), further separating MAOC into partly labile pool may hence be necessary for modelling, where this subfraction may strongly respond to changing DOC and necromass inputs under N addition.

#### 4.5 Outlooks

The recent review by Kuyper et al. (2024) and the study by Eastman et al. (2024) found a lack of soil models capable of explaining the diversity of C responses to N. In this study, we presented a process-based model that qualitatively aligned with multiple C responses to exogenous N addition. Despite the limitations, we gained significant ecological knowledge with our hypothesis-driven, incremental model variants. Our approach (similar to Zhang et al. (2020)) is different than common approaches such as whole-model comparisons and parameter tuning that encompass multiple confounding changes at once and are difficult to test ecological hypotheses and extrapolate model findings (Wieder et al., 2015b). We thus highly encourage future model studies to adopt a similar hypothesis-driven approach.

Although we did not explicitly represent microbial functional groups (e.g., fungi and bacteria) and their specialized enzymes, we based our models on stoichiometric variables: labile C:N and TER, since exoenzymes ultimately depend on available resources supply and demand (Sinsabaugh and Shah, 2012). This simplification greatly reduces the parameter requirement to explicitly describe various kinetics, growth, and allocation, community composition responses to nitrogen, where data are

scarce (Chandel et al., 2023; Moorhead et al., 2012; Moorhead & Sinsabaugh, 2006; Schimel & Weintraub, 2003; Wang et al., 2013). For instance, in a detailed microbial model, it is unclear whether we should model decomposition retardation as changes in C allocation to a specific enzyme, enzyme inactivation, microbial growth, or community composition (or their combinations), as data to disentangle the contribution of these co-occurring processes is lacking. Instead, stoichiometry defines biophysical limits (Buchkowski et al., 2015; Rocci et al., 2024; Zechmeister-Boltenstern et al., 2011), which could implicitly account for these underlying processes in aggregate. Our simple formulations thus permit easier implementation in any dynamic soil CN model under CMIP6 that calculates labile (readily-available) C:N and microbial biomass C:N.

Lastly, our model may be useful in other global change contexts (e.g., elevated CO<sub>2</sub> and warming) that entails changes in labile C and N availability. For instance, elevated CO<sub>2</sub> may increase labile C:N and intensify N-limitation, resulting in responses such as detrital decomposition retardation and microbial N mining (decomposition stimulation) from native low C:N SOM (Chen et al., 2014). Therefore, we encourage the use of our model (particularly the Nex<sub>retard</sub> + Nlim<sub>stim</sub> model containing microbial feedbacks up to dynamic CUE) for further testing and ecological applications.

## 5. Conclusion

Through our hypothesis-driven model experiment, we demonstrated that incorporating direct N effects on decomposer can reconcile model outputs with multiple observed responses of soil C pools qualitatively. Under N addition, models that incorporated the decomposition stimulation for N-limited substrates could reproduce a larger increase of LFH compared to topsoil SOC, as well as POC compared to MAOC commonly observed in experiments. Implementing dynamic, N-responsive microbial turnover drove microbial biomass reduction, while dynamic CUE maintained microbial C:N homeostasis to prevent erroneous estimations of N-limitation and excess. Based on these results, we proposed that N addition influences soil C dynamics primarily by speeding up high C:N detritus decomposition, while retarding the decomposition of processed OM with lower C:N ratios, as hypothesized. However, the intermediate pools POC and FH (the light fraction) showed the largest positive responses because they receive C directly from stimulated detritus decomposition, despite not having the lowest C:N ratio. Consequently, at contemporary levels of N deposition, we expected that most temperate forests will accumulate (or have accumulated) light fraction C predominantly, likely at the expense of high C:N detritus. Altogether, our model experiment provided robust mechanistic insights to soil N-C interaction, and we recommend our simple model for further testing and ecological applications.

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

### 1275 Supplements

The following supplementary materials are attached to this article:

Supplement 1: Additional figures and tables

Supplement 2: Full model and parameter documentation

## Code and data availability

The model source code (in C# code), model input files, validation data, and results analysis code (in R code) are all made available at Zenodo (https://doi.org/10.5281/zenodo.16953570).

### **Author contribution**

CCY is the first author and is the responsible author for all parts of the work including designing the study and writing the main texts and supplementary materials. HB and ODY are involved in the conception, planning and editing of the entire work. FH and MMD provided the Swiss forest soil validation data, offered scientific advice, and edited the main text.

#### Acknowledgements

This research cannot be finished without the support from 1) Swiss National Science Foundation whom provides the necessary research funding, 2) The Swiss Federal Institute for Forest, Snow and Landscape Research whom provides the relevant soil data, 3) The Swiss Federal Office for the Environment for providing the nitrogen deposition data, and 4) The Natural Resource Ecology Laboratory at the Colorado State University (especially Dr. Melannie Hartman) for providing the source code of the CENTURY model version 4.6. Additionally, we would like to thank all members of the Forest Ecology Group at ETH Zurich, and the first author would like to specially thank Gina Marano for the crucial support and advice on the use of the ForClim model.

# Financial support

This work is funded by the Swiss National Science Foundation (Grant No.: 188882).

# **Competing interests**

One of the authors is a member of the editorial board of Biogeosciences.