# Peer review of "How does nitrogen control soil organic matter turnover and composition? - Theory and model"

_EGUsphere, 2025_

## Author Comment (AC2)

**1. "*In this manuscript, Yeung et al. develop a new framework and model for the effect of N addition on soil organic matter cycling, accounting for both enhanced decomposition of N-poor substrates/soils and slower decomposition in N-rich soils and MAOM. The theory is tested using a model experiment, and the results compared to a meta-analysis of experimental responses to N addition. Overall, I think the manuscript will be an excellent contribution to the literature. The model experiment approach is compelling evidence for the hypothesized framework. The writing is very clear and the model is well-explained.*"**

We thank reviewer #1 for the positive evaluation of our manuscript.

**2. "*Major comment: How can the decomposition retardation framework (equations 3 to 5) can apply to both SOC2 (lignin-containing materials) and protein-rich MAOC? From the equations, it appears the effect is driven by the difference of the C:N of the pool compared to TER, but it seems like the C:N of SOC2 would nearly always be higher than TER. How would this simulate the observed inhibition of lignin-degrading enzymes?**

*Furthermore, the framework shown in figure 1 implies that decomposition lignin-rich but N-poor substrates like deadwood increase with N addition.*"

While it is true that the C:N of SOC2 is higher than MAOC and may experience less "N-excess" (hence less decomposition retardation), we wish to emphasize that not all lignin-containing OM (e.g., SOC2 and deadwood) is the same and experiences the same N effect. For example, the C:N ratio of SOC2 can vary from 12 to 57 in our model (see Supplement 2). In addition, we calculated the "labile C:N" (**eq. (4)**), not just the "C:N of SOC2" itself, to compare to TER and estimate the retardation (and CUE) effect. The labile C:N around SOC2, as well as litter and deadwood can be lower than TER (N-excess, retardation) according to our simulations under N addition (see **Fig. 4bgh**).

Since SOC2 (POC) covers a large C:N range (hence $labileCN_{POC}$), both a decomposition retardation and a stimulation response are possible (**Fig. 4b**). So, it is not contradictory that, as the reviewer suggested, SOC2 may not experience retardation if its C:N ratio is high, only some SOC2 did. The same goes for deadwood with its very high C:N ratio. Moreover, conceptually, deadwood lying on the forest floor and the organic matter in the H horizon differ in their C:N ratios and are also spatially separated, hence their lignin oxidases may have different responses to N. The multiple findings we highlighted in **lines 501-510** confirm these varying effects.

We understood the complexity of these concepts, and so we will make changes to clarify them (e.g., not all lignin-containing OM experiences the same N effect) in the sections *Introduction*, *Nitrogen-induced decomposer responses*, *Exogenous nitrogen effects on soil organic matter*. We will also modify the caption of **Figure 7** to highlight that the synoptic overview shows just the general response, and individual response may vary.

**3. "*Minor comments:**

*Line 159: "parameters" rather than "parameter"*

*Line 169: Suggest "turns over" instead of "turnovers"*

*Line 187: suggest adding "see section 2.6", as the meta-analysis approach has not been introduced at this point in the manuscript.*"

We will make all the suggested edits accordingly.

---

## Author Response (AR1)

The following document reported all the revisions and responses to the reviewer comments (including a community comment by Dr. Marijn van de Broek) received on manuscript egusphere-2025-1022. Line numbers correspond to the marked-up pdf version.

**Author responses to the comments by anonymous reviewer #1 on manuscript egusphere-2025-1022**

**1."In this manuscript, Yeung et al. develop a new framework and model for the effect of N addition on soil organic matter cycling, accounting for both enhanced decomposition of N-poor substrates/soils and slower decomposition in N-rich soils and MAOM. The theory is tested using a model experiment, and the results compared to a meta-analysis of experimental responses to N addition. Overall, I think the manuscript will be an excellent contribution to the literature. The model experiment approach is compelling evidence for the hypothesized framework. The writing is very clear and the model is well-explained."**

We thank reviewer #1 for the positive evaluation of our manuscript.

**#2.** "Major comment: How can the decomposition retardation framework (equations 3 to 5) can apply to both SOC2 (lignin-containing materials) and protein-rich MAOC? From the equations, it appears the effect is driven by the difference of the C:N of the pool compared to TER, but it seems like the C:N of SOC2 would nearly always be higher than TER. How would this simulate the observed inhibition of lignin-degrading enzymes?

Furthermore, the framework shown in figure 1 implies that decomposition lignin-rich but N-poor substrates like deadwood increase with N addition."

We agree that the C:N of SOC2 is higher than MAOC and may experience less "N-excess" (hence less decomposition retardation), but we wish to emphasize that not all lignin-containing OM (e.g., SOC2 and deadwood) is the same and experiences the same N effect. For example, the C:N ratio of SOC2 can vary from 12 to 57 in our model (see Supplement 2 Table S5). So, it is not contradictory that, as the reviewer suggested, SOC2 may not experience retardation if its C:N ratio is high, only some SOC2 did. The same goes for deadwood with its very high C:N ratio and hence it predominantly experienced decomposition stimulation by N. The multiple findings we highlighted in **lines 524-530** confirm these varying effects.

We have reflected the above points in the Introduction in the revised version (lines 57-58):

"Mechanistically, excess N suppresses the decomposition of some lignin-containing substrates (e.g., lignin enriched in particulate organic carbon – POC; Cotrufo & Lavallee, 2022),..."

And in Materials and Methods (lines 223-224):

"Note that lignin-containing substrates may experience either decomposition retardation eq. (3) or stimulation eq. (6), as lignin-containing substrates feature a wide range of C:N and hence labile C:N."

And in the Discussion (Line 562):

"Lastly, individual soil C responses varied with environmental factors despite the general pattern of response (cf. Table 2). Notably, <del>W</del>we found that cold soils..."

Lastly, in the caption of **Figure 7**: "A synoptic overview of the main decomposition effects of boundary N inputs on various organic matter pools. The illustrated effect sizes depict the general trends concluded from this study, but it must be noted that the effect varies for individual soils and organic matter that has a wide range of C:N ratios."

In addition, we wish to emphasize that our model calculates the "labile C:N" (eq. (4)), not just the "C:N of SOC2" itself to compare to TER and estimates the retardation (and CUE) effect. The labile C:N around SOC2, as well as litter and deadwood can be lower than TER (N-excess, retardation) according to our simulations with Nexretard models under N addition (see Fig. 4bgh). Labile C:N2 being one of the primary explanatory variables, was mentioned throughout the manuscript. We have added one more line to clarify this before eq (4), **line 199**: "Labile C:N surrounding C pool x is calculated by:"

Altogether, we believe that these changes sufficiently clarify that not all lignin-containing OM experiences the same N effect due to their wide range of C:N (and labile C:N).

**#3**. "Minor comments:

Line 159: "parameters" rather than "parameter"

Done.

Line 169: Suggest "turns over" instead of "turnovers"

Done.

Line 187: suggest adding "see section 2.6", as the meta-analysis approach has not been introduced at this point in the manuscript."

We appreciate the suggestion, but instead of referring to section 2.6, we have referred to "Table S5 in Supplement 2", which contains the correct references to both the numerical parameter values and the source meta-analyses.

**Author responses to the comments by anonymous reviewer #2 on manuscript egusphere-2025-1022**

We sincerely thank the reviewer for the thorough evaluation and the valuable comments.

In multiple instances the reviewer posed questions or requested us to address the limitations of the study (cf. section "*Limitations*" of the manuscript). We therefore provide the following summary of changes:

- 1. In the *Introduction*, we have clarified the scope and aim of our study i.e., modelling the general, decadal soil-C responses to N for potential integration into CMIP6 models. For this, we developed a stoichiometry-driven model, not a microbial model.
- 2. In the *Abstract, Introduction* and *Discussion*, we have strengthened the focus on stoichiometric process, e.g. N-limitation (Nlimstim) and stoichiometry-responsive CUE being critical for soil modelling.
- 3. We have performed additional analyses (e.g., separating temperate forest vs. global forest responses; probing the environmental dependencies of both observed and simulated response) to address the limitations, as explained in more detail below.
- 4. We have trimmed unnecessary content, removed non-essential conclusions (e.g., the decomposer response being more important than the plant response). We also limited the conclusions to temperate forests.

**Responses to specific comments**

**1. "In the Introduction part, what is the "state-of-the-art" soil model? Are they really "state-of-the-art"? How to define such a confusing concept? Additionally, the authors would be better to give some examples in detail to further explain such "state-of-the-art".**

We agree that the term "state-of-the-art" was confusing. Hence, 1) we have replaced "state-of-the-art models" by "CMIP6 soil models" as these are the models that are actually used for various soil C projections (see Arora et al., 2020; Spafford and MacDougall, 2021), and we have further clarified their current limitations. 2) We included the "state-of-the-art knowledge" on modelling N effects in general (not limited to CMIP6 models). These two changes have resulted in a major revision and reorganization in the *Introduction* lines 67-80):

"These decomposer responses have not been incorporated into CMIP6 soil models (Arora et al., 2020; Bouskill et al., 2014; Chen et al., 2019; Sun et al., 2016; Ťupek et al., 2016; Zaehle and Friend, 2010), despite the conclusion of multiple experimental studies that N addition exerted a larger effect on microbial decomposition than plant production. This is evidenced by decreased heterotrophic CO2 fluxes despite increased litter C inputs, and the predominant preservation of "old" C instead of a gain of "new" C in isotopic studies (Bowden et al., 2019; Franklin et al., 2003; Frey et al., 2014; Griepentrog et al., 2015; Hagedorn et al., 2003; Liu et al., 2024). Eastman et al. (2024) recently investigated the CASA (one of the soil models in CMIP6) and the microbial-explicit MIMICS models under N addition. The models reproduced most plant but only limited soil responses, despite including a post-hoc parameter adjustment representing decomposition retardation. The authors attributed the mismatch to a lack of relevant N-responsive microbial processes. A few modelling studies have also investigated decomposer response to N addition (Chen et al., 2019; Tonitto et al., 2014; Whittinghill et

al., 2012), but they again mainly featured a simple, static decomposition retardation factor on lignin decomposition, without considering other possible responses such as decomposition stimulation and microbial physiological changes dynamically (Fierer et al., 2012; Gravuer and Eskelinen, 2017; Hu et al., 2022; Knorr et al., 2005; Sinsabaugh et al., 2016). Here, we aim to fill this gap by creating a dynamic model embodying multiple N-induced decomposer responses."

**2. "In this article, the authors oversimplified the microbial dynamics. The model aggregates microbes into a single pool per soil layer, ignoring functional guilds (e.g., fungi vs. bacteria) that differ in N-use efficiency and substrate preferences. For instance, ectomycorrhizal fungi dominate lignin decomposition in boreal forests, but their response to N (e.g., suppression under N-excess) is not explicitly modeled. Additionally, it is about the TER (i.e., static TER). TER is calculated as microbial biomass C: N divided by maximum CUE, but this may oversimplify microbial adaptability. Recent studies have shown that TER can vary in response to nutrient stress and the dynamics of climate change, which could alter predictions of N-excess and limitation thresholds. I am curious about the model test scenarios where TER adapts to chronic N addition, and how this would affect estimates of N-excess vs. N-limitation?**

We acknowledge that our model simplifies microbial processes. However, we want to emphasize that no CMIP6 soil model currently features any detailed microbial dynamics, functional guilds or enzyme pools (due to limited detailed microbial data stretching long-term and wide environmental settings). The two microbial modelling studies under N addition that we know of (Aas et al., 2024; Eastman et al., 2024) both showed multiple mismatches with observations, and were analyzed rather phenomenologically. Generally, something being modelled explicitly does not mean it is modelled correctly, and something being modelled implicitly does not mean the model would fail to capture the underlying dynamic.

For our objective, we think that it is best to start from a simple null model lacking any relevant N response (i.e., CENTURY, also a parent model for many CMIP6 models; cf. **Figure 4**), rather than starting from a microbial model with pre-existing feedbacks and dependencies (with uncertain reliability in the context of N addition), which may then obscure an effect when we implement a new N-related process.

To clarify this, we have now emphasized the scope of application, the choice of a stoichiometric model over a microbial model, and the overall modelling objective in four places:

**(1) In Introduction after line 96:**

"...which may further impact decomposition (Bradford et al., 2017; Lladó et al., 2017; Treseder, 2008; Wu et al., 2023; Xu et al., 2021; process 5, Table 1). Noteworthy is that this stoichiometry-driven model is agnostic about detailed microbial dynamics (explicit changes in allocation to specific enzymes, enzyme kinetics, community composition and its functional traits, etc.). This is because field measurements are often aggregate outcomes of co-occurring microbial processes that cannot be disentangled (e.g., aggregate element flow, which is in the domain of stoichiometry). We aim to capture the general decadal scale soil C responses to N that could potentially be incorporated into CMIP6 models for predictions in the near-term."

**(2) In **lines 110-111**:**

"Based on this conceptual model and our aim, we developed equations capturing these five N-induced decomposer responses, which were implemented in a benchmark base model derived from CENTURY that has no N-induced decomposer response"

**(3) In **lines 478-484**:**

"Previous studies typically adopted a simple parameter adjustment to represent a one-time, static response to N in a process deficient model decomposition retardation effect (Chen et al., 2019; Eastman et al., 2024; Tonitto et al., 2014; Whittinghill et al., 2012), or derived theoretical ecoenzymatic models responsive to nutrients, which contain many detailed processes and parameters that are difficult to estimate or constrain (Moorhead & Sinsabaugh, 2006; Schimel & Weintraub, 2003; Sinsabaugh & Shah, 2012; Wutzler et al., 2017). Our models represent a compromise between the two with the use of simple stoichiometric principles, taking into account the data available for use to develop models."

**(4) In the section "Outlook" lines 659-669:**

"Although we did not explicitly represent microbial functional groups (e.g., fungi and bacteria) and their specialized enzymes, we based our models on stoichiometric variables: labile C:N and TER, since exoenzymes ultimately depend on available resources supply and demand (Sinsabaugh and Shah, 2012). This simplification greatly reduces the parameter requirement to explicitly describe various kinetics, growth, and allocation, community composition responses to nitrogen, where data are scarce (Chandel et al., 2023; Moorhead et al., 2012; Moorhead & Sinsabaugh, 2006; Schimel & Weintraub, 2003; Wang et al., 2013). For instance, in a detailed microbial model, it is unclear whether we should model decomposition retardation as changes in C allocation to a specific enzyme, enzyme inactivation, microbial growth, or community composition (or their combinations), as data to disentangle the contribution of these co-occurring processes is lacking. Instead, stoichiometry defines biophysical limits (Buchkowski et al., 2015; Rocci et al., 2024; Zechmeister-Boltenstern et al., 2011), which could implicitly account for these underlying processes in aggregate. Our simple formulations thus permit easier implementation in any dynamic soil CN model under CMIP6 that calculates labile (readily-available) C:N and microbial biomass C:N."

Also in the Abstract (lines 18-19 and line 29):

"Our results support the hypothesis that N directly influences multiple C pools by changing decomposition and microbial physiology, which are in turn driven by stoichiometric imbalances."

"Collectively, our model experiment provided robust mechanistic insights on the stoichiometric control of soil N-C interaction."

Regarding ectomycorrhizal fungi and TER, we emphasize that TER in our model is dynamic (it varies with microbial biomass C:N, e.g., ectomycorrhiza are more common in colder sites and have a relatively higher biomass C:N and hence TER), and we were able to capture the higher tendency of decomposition retardation in colder (similar to boreal) sites, as the reviewer expected (see **Table 2** and discussed in **lines 563-573**). In contrast, using a modified MIMICS (Aas et al., 2024), a model with explicit functional groups, ectomycorrhizal fungi had almost no response and there was no decomposition retardation under N addition.

To clarify this, we have added in **lines 569-570**:

"... resulting in a stronger decomposition retardation of recalcitrant substrates (Kuyper et al., 2024; Lilleskov et al., 2019; Zhou et al., 2017). Although microbial functional groups are not explicitly represented in our models, oligotrophic decomposers in cold soils correlate with a higher biomass C:N and TER, which led to a higher chance of N-excess in our model. In contrast..."

We have stated in **lines 86-87** and **lines 210-211** why our TER definition "MBCN / CUEmax" is appropriate (see also Doi et al., 2010; Sinsabaugh et al., 2013). We modified **lines 86-87** slightly to make the reasoning clearer: "We then considered the Threshold Elemental Ratio (TER; Mooshammer et al., 2014a; Sinsabaugh et al., 2013), defined as the ratio of readily-available C and N resources for decomposers at which no nutrient C and N limitations and optimal decomposition occur."

Considering a changing CUE and NUE in the definition of TER (as the reviewer pointed out) is only suitable as a measure of bulk substrate C:N demand, not labile (readily-available) C:N. An example is that of microbes decomposing litter of high C:N ratio (cf. Manzoni et al., 2010). In this case, reduced CUE is an "adaptation" that increases "TER", but this "TER" is a measure of *bulk* litter C:N, and if CUE has to be adjusted, it is no longer *optimal* per se (lower CUE -> less C to biomass and enzymes, which leads to "suboptimal" decomposition. cf. **Fig. 1**).

To make the rationales of our stoichiometric equations clearer, we have further provided a complete documentation of the choice of variables, parameters and function forms of all the new equations in **Supplement 2 sect. 5**.

**3. We place our responses to comments #3 and #7 together, since they both addressed MAOC (see response to #7 further below).**

**4. "While the model tests MAT and MAP effects, it does not explicitly link temperature to microbial enzyme kinetics or N mineralization rates. For example, in warm forests, N addition may accelerate the mineralization of labile C through temperature-microbial interactions, which are not captured here. Contemporary N deposition data are assumed to be representative of long-term trends; however, historical peaks (e.g., the 1980s in Europe) may have legacy effects on SOC that are not reflected in the model. Additionally, while the Swiss forest dataset is robust, testing the model in other biomes (e.g., boreal, tropical forests) would validate its generality. For example, N-limitation is more pronounced in tropical soils, potentially amplifying decomposition stimulation effects on high C:N litter. So, in tropical forests with high C:N litter and strong N limitation, would decomposition stimulation under N addition be more pronounced than simulated?"**

Thanks for the comments. We agree that the different processes discussed in this comment could be relevant, and we streamlined our response as follows:

**1. Temperature-microbe interactions**

Our model includes a temperature effect on C and N mineralization inherited from CENTURY (eq. S9), which is widely used and implicitly accounts for the temperature dependency of enzyme kinetics. A more explicit treatment of temperature—enzyme relationship is beyond the scope of our manuscript, as we focus on N effects. We emphasize the importance of staying in focus, as the manuscript is already long, and adding extra things may distract readers from understanding.

**2. Legacy effect of historical N deposition**

In the original manuscript, we did mention a possible underestimated contemporary N deposition (~6-38 kgN ha-1 y-1), which may make the N effect less prominent as compared to N-addition experiments (40-150 kgN ha-1 y-1). Nonetheless, we used N deposition estimates averaged over the 1990s till 2010,

which are not much lower than in the 1980s on closer inspection, and are at comparable levels to 1940-70s deposition (cf. Gharun et al., 2021 and Schöpp et al., 2003).

As stated, contemporary N deposition effect tended to be small (not counting in concurrent drivers such as sulphur acid deposition). Meta-analyses of N addition experiments confirmed that at low N addition dosages, all the effects on soil C pools (and plant C pools) were generally smaller (cf. revised text below). It is unlikely there were strong legacy effects driven solely by the slightly higher N deposition around 1980s briefly.

In the revised manuscripts, we have clarified the above and rewrite the paragraph from lines 595-600:

"Overall, these findings under contemporary N conditions tend to converge with the N addition findings prior (e.g., larger light fraction C accumulation, which is captured by the Nexretard + Nlimstim models). However, not surprisingly, the differences between model variants were smaller, as N deposition is typically smaller (6-38 kgN ha-1 y-1) than the amount of N added in experiments (40-150 kgN ha-1 y-1). Multiple meta-analyses have found dose-dependent responses of both soil and plant variables, where low N addition (<50 kgN ha-1 y-1) tended to produce small effects not different from controls (Chen et al., 2015; Jian et al., 2016; Liu et al., 2024; Tang et al., 2023; Xu et al., 2021)..."

In addition, in **lines 292-294**, we have changed: "We acknowledge that tThese N deposition data are representative of the 1990s and 2000s when N deposition was on decline and hence were likely underestimated and comparable to the level in 1940-1970s, with the exception of 1980s when deposition peaked (cf. Gharun et al., 2021; Schöpp et al., 2003)."

In the Abstract (lines 25-26), we have added: "... This concurrently explains the organic horizon and POC accumulation under contemporary N deposition in temperate forests, albeit with smaller effect sizes than in N addition experiments where N application rates are higher."

Moreover, to show that our conclusion is robust even if we increase N deposition (as the reviewer suggested), we have conducted a sensitivity test with a 20% increase of contemporary N deposition, and have re-run the contemporary simulation analysis (i.e., Fig. 6 and Table S4). We found that again the Nexretard + Nlim $_{\text{stim}}$  models with the dynamic CUE and microbial biomass feedback enabled matched the closest to observed trend by far (p >> 0.05 statistically not different from the observed slope; echoing lines 454-459). Since the general result remains the same, we do not intend to include this sensitivity test result in the manuscript.

**3. Evaluation across biomes**

We agree with the reviewer that expanding the simulation to boreal and tropical sites would be interesting. However, this would require appropriate input and evaluation data from other biomes, which are quite limited. As an alternative, we have conducted addition analyses: 1) multiple regression of MAT, MAP, and clay on the observed meta-analysis responses (same as what we did for simulated response in Table 2); 2) Re-plotted **Figure 4** (cf. below) to distinguish temperate vs. global forest sites (with added data from Xu et al. (2021) to improve the generality of LFH response). We only separated the biomes into temperate forests as the number of datapoints are quite low for other biomes. From these analyses, we have found:

- 1) The observed LFH, SOC and MAOC responses did positively correlate strongly with MAT (new Table S3), same as the simulated responses of the  $Nex_{retard} + Nlim_{stim}$  models (Table 2). This is in line with the reviewer's expectation, and we had already discussed this in the manuscript (lines 563-573). We have reported these new results in lines 432 435:
- "Climatic effects were notable as MAT (warmer) and MAP (wetter) showed significant positive effects on the simulated response of LFH, SOC, POC, MAOC, MBC and Rh, but a negative effect on deadwood. This aligns with the higher observed responses of LFH, SOC, MAOC in warmer climate (Table S3). The observed LFH response correlates negatively with MAP in temperate forests, which did not align with any model."
- 2) The general response patterns of temperate forests align with global forests (**new Figure 4**), supporting the model's broader applicability. Following this **new Figure 4**, we have only made minor edits to the results from **line 391-402**. As mentioned, we also limited our conclusion to temperate forests due to the smaller number of datapoints in other biomes, so as to not over-interpret.
- 3) In addition, in the *Discussion*, we encouraged future studies and more data collection in other biomes in **lines 605-607**:
- "However, we found a strong consistency in the general response pattern regardless of the forest biome classification (cf. Fig. 4. See also Ramirez et al., 2012). Nonetheless, we urge more experiments to be conducted outside the temperate zone where data is still limited..."

Table S3. Multiple-regression of the site factors that control meta-analysis percent response of selected C-cycle variables under N addition. Standardized coefficients are scaled by their standard deviation.

| LFH response    | Global forests   |         | Temperate forests*   |         |
|-----------------|------------------|---------|----------------------|---------|
| Predictors      | Std. Coefficient | P-value | Std. Coefficient     | P-value |
| Intercept       | NA               | 0.242   | NA                   | 0.302   |
| MAT             | 0.58             | 0.094.  | 0.98                 | 0.008** |
| MAP             | -0.62            | 0.071.  | -0.84                | 0.018*  |
| Cum. N load †   | 0.03             | 0.909   | 0.15                 | 0.594   |
| N addition rate | 0.22             | 0.409   | 0.13                 | 0.626   |
| SOC response    | Global forests   |         | Temperate forests    |         |
| Predictors      | Std. Coefficient | P-value | Std. Coefficient     | P-value |
| Intercept       | NA               | 0.205   | NA                   | 0.665   |
| MAT             | 0.05             | 0.852   | 0.79                 | 0.014*  |
| MAP             | -0.03            | 0.904   | -0.1                 | 0.686   |
| Cum. N load †   | -0.02            | 0.925   | 0.38                 | 0.197   |
| N addition rate | -0.12            | 0.575   | -0.51                | 0.073.  |
| Clay            | -0.06            | 0.745   | -0.14                | 0.536   |
| POC response    | Global forests   |         | Temperate forests    |         |
| Predictors      | Std. Coefficient | P-value | Std. Coefficient     | P-value |
| Intercept       | NA               | 0.538   | NA                   | 0.195   |
| MAT             | -0.39            | 0.169   | -0.2                 | 0.502   |
| MAP             | 0.44             | 0.11    | 0.28                 | 0.277   |
| Cum. N load †   | -0.34            | 0.087.  | -0.25                | 0.398   |
| N addition rate | 0.37             | 0.1     | 0.56                 | 0.058.  |
| Clay            | -0.01            | 0.954   | 0.17                 | 0.434   |
| MAOC response   | Global forests   |         | Temperate forests*** |         |
| Predictors      | Std. Coefficient | P-value | Std. Coefficient     | P-value |
| Intercept       | NA               | 0.028*  | NA                   | 0.085.  |
| MAT             | 0.84             | 0.059.  | 0.64                 | 0.122   |
| MAP             | -0.72            | 0.07.   | -0.39                | 0.145   |
| Cum. N load †   | 0.27             | 0.282   | -0.38                | 0.469   |
| N addition rate | -0.23            | 0.397   | 0.4                  | 0.422   |
| Clay            | -0.29            | 0.13    | -0.38                | 0.016*  |
| Rh response     | Global forests   |         | Temperate forests    |         |
| Predictors      | Std. Coefficient | P-value | Std. Coefficient     | P-value |
| Intercept       | NA               | 0.265   | NA                   | 0.112   |
| MAT             | -0.42            | 0.198   | 0.09                 | 0.781   |
| MAP             | 0.28             | 0.346   | 0.76                 | 0.057.  |
| Cum. N load †   | -0.32            | 0.369   | 0.83                 | 0.372   |
| N addition rate | 0.59             | 0.063.  | -0.78                | 0.414   |

\* For P-values, one asterisk indicates p < 0.05, two asterisks indicate p < 0.01, three asterisks indicate p < 0.001.

† CumN load is the cumulative N addition calculated by N addition rate (y-1) times experimental duration (y)

Figure 1. Global observed responses and simulated responses of 54 Swiss forest sites under N addition: % difference in N treatment vs. control of (a) LFH C stock, (b) mineral soil POC stock, (c) mineral soil SOC stock, (d) mineral soil MAOC stock, (e) MBC (surface + mineral soil), (f) MBC:MBN (surface + mineral soil), (g) surface fresh litter C stock, (h) deadwood C stock, (i) annual heterotrophic respiration (surface + mineral soil), and (j) total dead organic C stocks in the base model and eight model variants averaged over the year 5-15 after the start of N addition. The solid orange vertical lines are the observed global meta-analysis means, and the dot-dash lines are the lower and upper quartile values. The green shaded region consists of a subset of temperate forests only (by IGBP-DIS classification (Woodward et al., 2004). N is the global forest sample size and n is the temperate forest.

**5. "By excluding plant responses (e.g., root exudation, litter quality changes), the model may underestimate N effects in systems where plants dominate C input shifts. For instance, N-induced shifts from fine roots to aboveground litter could alter SOC pool dynamics, a pathway not explored here. Would incorporating plant-derived changes in litter quality (e.g., lower lignin: C ratios) alter the predicted dominance of microbial vs. plant-driven SOC changes? Fine root decomposition contributes to POC and MAOC, but the model treats all litter inputs uniformly. How does the lack of root-soil carbon pathway distinction affect predictions of N-induced POC accumulation?"**

We agree that N-driven changes in plant inputs (e.g., root exudation, litter quality shifts, fine-root turnover) can influence SOC dynamics, but these processes fall outside our decomposer-focused modelling objectives. Addressing them as part of the model variants would be excessively complex (note that none of the earlier modelling studies had addressed more than three processes), hence we can only discuss them. Accordingly, we have made the following changes:

- removed all instances where we claimed that our findings showed that the decomposer response is more important than the plant response;
- expanded the discussion (through **lines 615–627**) to acknowledge that increased aboveground litterfall may be important. In contrast, variable fine-root production/turnover response (ranging from decrease to increase) under N addition warrants more investigation. A lack of empirically

robust, general relationships is the primary reason of the difficulty to include it in a model currently;

- recommended that a logical next research should focus on plant responses
- further discussed possible interactions between the plant and decomposer responses (lines 615-627):

"First, N addition generally increases aboveground litterfall, which would increase and potentially improve the LFH response (Chen et al., 2015; Janssens et al., 2010). However, litterfall alone may not explain the large LFH increase. Several studies (Bowden et al., 2019; Frey et al., 2014; Lovett et al., 2013) reported no significant change in aboveground litterfall yet observed substantial LFH increases. For example, Bowden et al. (2019) observed an 80% increase in the organic horizon of a mature hardwood forest after 17 years of N addition, which could be attributed neither to litterfall (no significant change) nor to suppressed soil respiration (only -5.7%). Schulte-Uebbing and de Vries (2018) reported that increased aboveground production was mainly associated with young stands where experiments are more common, and hence more studies should be conducted in mature forests. Root litter response ranged from a decrease to an increase, leading to uncertain effects on both FH and SOC stocks (Chen et al., 2015; Feng et al., 2023; Franklin et al., 2003; Frey et al., 2014; Peng et al., 2017; Yue et al., 2021). Again, this highlights the need for more studies on how N influences root litter production and their modulating factors. Once these are better quantified, incorporating the litter response into new models is highly valuable (Xia et al., 2018), as the Nlimstim effect by definition interacts with the amount of litter inputs, where more litter may amplify the role of N-limitation."

**6. "Nitrogen Excess vs. Acidification??? N addition and acidification often co-occur, but the model attributes decomposition retardation to N-excess alone. How were the effects of N-induced acidification disentangled from stoichiometric imbalances, especially in acidic Swiss forests (pH range is not narrow, instead, it is large)?"**

We addressed in **lines 629-640** why we think nitrogen, not acidity, control the bulk of responses in our study. To further support our argument, we have included a recent study by Frey et al. (2025) that made use of both N manipulation and pH manipulation experiments, showing the same conclusion as us, i.e. that nitrogen, not acidity, mainly controls the soil C response under N addition. We have added this information between **lines 632-633**:

"Nonetheless, acidification alone cannot explain any process other than decomposition retardation (e.g., stimulation). Even for retardation effect, Frey et al. (2025) showed that it was mainly driven by increasing N availability in a field experiment designed to disentangle N and acidification effects. Besides, studies had shown that nitrogen and acidification led to different microbial community responses..."

**3. "The MAOC turnover and pH effect are uncertain in the model discussion. For example, MAOC is assumed to be a homogeneous "slow-turnover" pool, yet real-world MAOC contains labile sub-fractions, which could explain the model's short-term MAOC response lag. Additionally, The base model includes pH retardation on decomposition but assumes uniform pH across soil layers, contradicting evidence that FH horizons are more acidic than mineral soil. This may overestimate LFH accumulation in acidic forest soils."**

**7. "The model evaluates N-addition effects over 5–15 years, aligning with most experimental durations, but meta-analyses show divergent long-term trends (e.g., MAOC accumulation). Given that microbial necromass and mineral-associated C turnover occur over decades to centuries, how does the model validate its precision in predicting transient vs. asymptotic SOC responses, especially for slow-turning pools like MAOC, and are there plans to incorporate time-dependent microbial adaptation (e.g., enzyme induction/suppression) to improve long-term predictability? Can the model's long-term MAOC predictions be validated using centennial-scale soil carbon datasets?"**

Comment #3 referred to two limitations – uniform pH effect and homogeneous MAOC representation. The latter contributes to the uncertainty in the transient MAOC response in the model (comment#7). In addition, the validation of asymptotic (equilibrium) response is not possible currently for slow-turnover pools like MAOC. Since these are rather complex issues, we broke down our response in the following sections:

**1. pH effects:**

Modelling pH effect is under-studied (mentioned in **line 496**). However, fixing this would be a whole other study (which we are investigating in a separate work). To show our conclusion is still robust, we have re-plotted **Figure 4** without acidic sites (pH < 4, where the modelled pH effect only starts to become important). Our conclusion still holds: Nexretard + Nlimstim models worked better and other model variants produced various responses far outside reasonable ranges. The removed acidic sites generally featured overestimated C stocks, which reduced the magnitude of percent response and so removing them actually improved the pattern of response (especially for Nexretard + Nlimstim models, see **Table 2**). This figure is added in Supplement 1 (in text reference as **Fig. S7**) and described in **lines 506-510:** "...but it overestimated topsoil SOC primarily under acidic conditions (irrespective of nitrogen effects). Nonetheless, we emphasize that our focus is the response of various C pools under changing N availability, and removing acidic sites did not affect the general response pattern (Fig. S7). In fact, overestimated C stocks contributed to smaller percent responses especially for Nexretard + Nlimstim models (Table2), so their pattern of response improved when the overestimated sites were removed."

Ι

Figure S7. Same as Figure 4, but with acidic sites (topsoil pH < 4) removed.

**2. MAOC representation in models and its transient response:**

- a) We have expanded the discussion (in **lines 641–650**) about modeled MAOC being treated as a homogeneous "slow" pool, whereas several studies identified a further sub-division of a relatively labile (more-responsive) vs. resistant pools (Guo et al., 2022; Poeplau et al., 2018; Schrumpf and Kaiser, 2015). This lack of finer detail (a common problem in most soil models, not limited to CENTURY) may contribute directly to the failure to capture the large variation of MAOC transient response, as the reviewer noted in comment #7.
- b) We have further added recent results from our Swiss forest dataset (**lines 646-647**), where we also sub-divided MAOC into oxidizable (32–1440 y) and resistant (130–4597 y) fractions. The paragraph (**lines 641–650**) have been re-written to include this and its interpretations:

"Another uncertainty is the lack of MAOC response in our N addition simulations (Fig. 4D). However, MAOC did respond on a longer time scale in the Nexretard + Nlimstim models (cf. Fig. S5), matching the mean response in temperate forests. Nonetheless, the large variation of meta-analysis MAOC response could not be captured. A possible reason is that experimentally fractionated MAOC contains sub-fractions of labile, fast-turnover compounds not represented by the modelled "homogeneously-stable" MAOC (Brunmayr et al., 2024; Guo et al., 2022; Poeplau et al., 2018; Schrumpf and Kaiser, 2015; Sokol et al., 2022). For Swiss forests, Moreno-Duborgel et al. (2025) subdivided MAOC further into a peroxide-oxidizable and residual (resistant) fractions and showed distinct turnover times of 32–1440 and 130–4597 y, respectively. The turnover times of MAOC also vary considerably from decades to millennia in the literature (Guo et al., 2022; Kleber et al., 2015; Lavallee et al., 2020), further separating MAOC into partly labile pool may hence be necessary for modelling, where this sub-fraction may strongly respond to changing DOC and necromass inputs under N addition."

**3. Asymptotic response and the controls of MAOC pool in the long-term:**

- a) We have highlighted that our objective is "general, decadal soil C responses to N that could potentially be incorporated into CMIP6 models for predictions in the near-term" as stated in answer #2. A direct validation at the centennial-millennial scale is currently not possible due to the absence of long-term N manipulation experiment data, or repeated fractionated SOC inventory data dating to pre-1940s era (before N deposition intensifies).
- b) To improve the understanding on the long-term control of MAOC via N-related mechanisms, we have revised (lines 584-594) substantially in the discussion of contemporary N effect:

"However, this negative trend disappears when considering the relationship between MAOC:N and MAOC:total litter C input — an indicator of the overall conversion efficiency of detrital C to MAOC and its persistence. The base model and simple Nexretard models clearly overestimated MAOC conversion/protection at low MAOC:N and falsely retained the negative trend. This was substantially alleviated in the Nexretard + Nlimstim models with microbial feedback enabled, again highlighting their importance and that retardation effect cannot be solely considered. Essentially, enabling N limitation and microbial feedback limits OM processing, thereby reducing the conversion to MAOC. Nonetheless, no model can capture MAOC stocks in high MAOC:N soils (coinciding cold, coniferous, acidic soils). Several studies found that leaching of dissolved organic C from coniferous forest floors throughout the year (not just in autumn) was conducive to MAOC formation (Andivia et al., 2016; Bramble et al., 2024; Córdova et al., 2018). Our models do not contain dissolved C dynamics, which may imply a need for further model improvement as fresh litter decomposition and hence leaching from the litter layer is likely enhanced by N (Hagedorn et al., 2012b)."

**References (only new references that are not in the current manuscript text)**

[revised manuscript text omitted]

**Author responses to the comments by community reviewer on manuscript egusphere-2025-1022**

We thank Dr. Marijn van de Broek for the thorough evaluation of our manuscript and the numerous valuable suggestions.

1. "My main feedback relates to the choice of the model (CENTURY). The structure of this model is based on the humification theory, which has been criticised in the literature (Lehmann and Kleber, 2015 (doi.org/10.1038/nature16069); Kleber and Lehmann, 2019 (doi.org/10.2134/jeq2019.01.0036)) and is considered outdated. Although the names of the pools have been changed to measurable pools, as was done in previous studies, the structure is different from the general understanding of SOM cycling (e.g., there is no DOC pool, there is a direct flux of POC to MAOC, although in reality POC is depolymerised to DOC first, which can then either be adsorbed on minerals, taken up by microbes or leached from the soil, and the amount of soil microbes does not affect the rate of decomposition (although this is implemented as a scenario in the study)). However, the authors consistently refer to this model as 'process-based'. Over the past two decades, multiple models simulating coupled soil C - N cycles using measurable pools, while explicitly simulating established mechanisms which are addressed in the present study (e.g., adsorption and desorption of DOC on/from soil minerals, the effect of microbial biomass (or enzymes) on rates of polymerisation and decomposition, microbial CUE, etc.). Therefore, I would suggest the authors to better justify their choice for this model, over other, more mechanistic, SOM models, as these explicitly incorporate many of the simulated mechanisms related to soil microbes and MAOC discussed in the present manuscript.

We thank Marijn for the comment on the choice of the (baseline) CENTURY model.

Firstly, we used CENTURY as a "benchmark" (see **Fig. 4**, where it has almost no response to N), as has been done in many soil model studies to show the improvement of a new model. Our relatively general equations (**eq. 3 to 11**) are not necessarily an extension of CENTURY, but could be implemented in other (CMIP6) soil models for instance (**lines 668-669**). Secondly, the choice of model depends on the objective of the study, and the data available to develop and test a model. Currently, mainstream CMIP6 soil models do not contain any detailed microbial processes and hence we use CENTURY for an easier integration. To highlight the modelling objective more clearly, we have added after **line 96**:

"...which may further impact decomposition (Bradford et al., 2017; Lladó et al., 2017; Treseder, 2008; Wu et al., 2023; Xu et al., 2021; process 5, Table 1). Noteworthy is that this stoichiometry-driven model is agnostic about detailed microbial dynamics (explicit changes in allocation to specific enzymes, enzyme kinetics, community composition and its functional traits, etc.). This is because field measurements are often aggregate outcomes of co-occurring microbial processes that cannot be disentangled (e.g., aggregate element flow, which is in the domain of stoichiometry). We aim to capture the general decadal scale soil C responses to N that could potentially be incorporated into CMIP6 models for predictions in the near-term."

In addition, we have re-written lines 109-110:

"Based on this conceptual model and our aim, we developed equations capturing these five N-induced decomposer responses, which were implemented in a benchmark base model derived from CENTURY that has no N-induced decomposer response"

Thirdly, we respectfully disagree that CENTURY v4.6 embodies humification theory. In CENTURY, the flow into slow-turnover SOC3 (MAOC) depends on clay and is always accompanied by a concurrent flow to an unprotected pool (**Figure 2**), similar to the clay-dependency of sorption many newer models employ (with the remainder staying unprotected). The difference is only that dissolved organic carbon (DOC) as an intermediate is skipped, which is justifiable in a model with a monthly time step, as data are often not available to validate DOC. Also, we did not have a hypothesis about N effect on DOC. Models like RothC, YASSO and ICBM are closer matches to humification theory.

2. "Related to this, in L462, the authors cite studies using models which are very similar (Chen et al., 2019; CASA CNP in Eastman et al., 2024), or more mechanistic (MIMICS in Eastman et al., 2024) to the model they used, and refer to these as 'process-deficient'. How do the authors reconcile this with referring to CENTURY as a mechanistic model? In the same sentence, they refer to models containing multiple mechanisms as containing 'many uncertain parameters and hence [it is] difficult to verify the relevance of each process'. However, in their study the authors use mainly default parameters for the CENTURY and FORCENT models. I would encourage the authors to explain why the parameter values in the latter would be less uncertain compared to those in state-of-the-art mechanistic SOM models."

"L461-465: instead of dismissing previous modelling studies as 'process-deficient' or 'containing many uncertain parameters', it would have been worthwhile to summarize their findings in the introduction as a background to the modelling study that is presented here."

**Author response:**

We used "Process-based" in reference to our newly implemented N-effect model (in the same vein, "Process-deficient" is referring to a lack of N effect process, not in general about the whole model). To avoid confusion, we have deleted in **line 479:** "Previous studies typically adopted a simple parameter adjustment to represent a one-time, static response to N in a process deficient model decomposition retardation effect"

We stand by our discussion that detailed enzymatic (microbial) models contain many uncertain parameters and processes. Taking the example of the N retardation effect on lignin decomposition, using a mechanistic microbial model would raise the issue whether this should be modelled as changes in C allocation to (a specific) enzyme, microbial community, microbial uptake and growth, enzyme inactivation, or their combination, etc. For none of these mechanisms do we currently have reliable empirical data. As stated in answer #1, we added a clarification after **line 96**. In addition, in "Outlook" **lines 664-669**, we have re-written:

"[...] For instance, in a detailed microbial model, it is unclear whether we should model decomposition retardation as changes in C allocation to a specific enzyme, enzyme inactivation, microbial growth, or community composition (or their combinations), as data to disentangle the contribution of these co-occurring processes is lacking. Instead, stoichiometry defines biophysical limits (Buchkowski et al., 2015; Rocci et al., 2024; Zechmeister-Boltenstern et al., 2011), which could implicitly account for these underlying processes in aggregate. Our simple formulations thus permit easier implementation in any dynamic soil CN model under CMIP6 that calculates labile (readily-available) C:N and microbial biomass C:N."

The limitations of detailed microbial models with uncertain parameters and parameter equifinality issues is well-documented outside our study (Brunmayr et al., 2024; Luo et al., 2009; Marschmann et

al., 2019; Schimel and Weintraub, 2003; Van de Broek et al., 2025). Since this is not a manuscript about microbial vs. non-microbial modelling, further in-depth discussion is out of the scope.

3. "In their study, the authors compare simulated effects of N addition to Swiss forest soils with the results from a global meta-analysis. I would encourage the authors to better (graphically) represent the differences in environmental characteristics between the Swiss sites and the sites from the meta-analyses. Related to this, the authors justify the comparison to results from global meta-analyses by stating that 'most experiments were conducted in temperate forests' (L330). Which portion is 'most'? It would be good to justify why not only these experiments from temperate forests were used in the comparison, instead of also using experiments from other ecosystems."

"L318-327: as the data was extracted from global meta-analyses, it would be good to report, for example, from which ecosystems these data were collected. In addition to table 2, the authors can consider a graphical representation of the environmental characteristics of the sites from the meta-analyses, overlain by the characteristics for the Swiss forests. This information is essential for the reader to evaluate the comparison that is made with the Swiss forest sites. Related to this, on L575 the authors state the 'environmental gradient coverage [of the Swiss sites] differs \*slightly\*' from the sites in the meta-analyses', while, for example, MAT for the Swiss sites ranges from 6.5 to 9.4 °C, while this is between 1.0 and 21 °C for the sites from the meta-analyses. This statement thus seems misleading, and a better representation of these differences seems necessary to inform the reader, as the comparison between the simulation results obtained from the Swiss sites and those from the meta-analyses is at the core of this study."

We fully acknowledge this limitation and have conducted additional analyses to address it. We have re-worked **Figure 4** (cf. below) to distinguish temperate vs. global forest sites (plus further increasing the LFH dataset from n=20 to 28 with data from Xu et al. (2021)). The general response patterns of temperate forests align closely with global forests, supporting the model's broader applicability. Following this **new Figure 4**, we made minor edits to the results from **line 391-402**. Moreover, we provided new frequency distributions of MAT and MAP (these are the only two environmental attributes consistently available across the multiple meta-analyses) to be included in *Supplement 1* as **Fig. S3**. Lastly, we have added in **lines 605-607**:

"...However, we found a strong consistency in the general response pattern regardless of the forest biome classification (cf. Fig. 4. See also Ramirez et al., 2012). Nonetheless, we urge more experiments to be conducted outside the temperate zone where data is still limited..."

Figure 4. Global forest observed responses and simulated responses of 54 Swiss forest sites under N addition: % difference in N treatment vs. control of (a) LFH C stock, (b) mineral soil POC stock, (c) mineral soil SOC stock, (d) mineral soil MAOC stock, (e) MBC (surface + mineral soil), (f) MBC:MBN (surface + mineral soil), (g) surface fresh litter C stock, (h) deadwood C stock, (i) annual heterotrophic respiration (surface + mineral soil), and (j) total dead organic C stocks in the base model and eight model variants averaged over the year 5-15 after the start of N addition. The solid orange vertical lines are the observed global meta-analysis means, and the dot-dash lines are the lower and upper quartile values. The green shaded region consists of a subset of temperate forests only (by IGBP-DIS classification (Woodward et al., 2004). N is the global forest sample size and n is the temperate forest.

Figure S3. Relative density distributions of the occurrence of mean annual temperature (MAT) and mean annual

precipitation (MAP) in all of the global forest meta-analyses used in this study, along with the Swiss forest environmental gradient.

4. "L31-32 and L637-638: You state that your result challenge the 'common assumption that plant is the primary respondent to N'. This is a strong statement, as you (from what I understood in the manuscript) did not simulate the response of plants to N addition, or interactions between the response of plants and soil microbes. Therefore, it makes sense that without simulating the response of plants, the response of soils will be large. However, as the response of plants wasn't quantified, I suggest to revise this statement."

We appreciate this comment, as it pointed out an 'overselling' of our results. We have removed all instances where we claimed that our findings showed that the decomposer response is more important than the plant response.

5. "L52: you state that 'Efforts to synthesize the diverse responses into a coherent theory and model are lacking', although many modelling studies have studied this effect, of which you cite a few. Therefore, I encourage the authors to modify this statement. In addition, as this is a modelling study, it would have been useful to summarise the outcomes of these previous modelling studies with respect to the response of soil organic matter characteristics to N addition."

Thanks for the comment. We have highlighted more strongly the "state-of-the-art knowledge" on modelling N effects in the *Introduction*. However, we do not think that "many modelling studies have studied this effect" comprehensively. We only know of Chen et al., 2019, Tonitto et al., 2014, Whittinghill et al., 2012, Eastman et al., 2024 at the time of writing the manuscript. We have included the additional information below (lines 72-80):

"...Eastman et al. (2024) recently investigated the CASA (one of the soil models in CMIP6) and the microbial-explicit MIMICS models under N addition. The models reproduced most plant but only limited soil responses, despite including a post-hoc parameter adjustment representing decomposition retardation. The authors attributed the mismatch to a lack of relevant N-responsive microbial processes. A few modelling studies have also investigated decomposer response to N addition (Chen et al., 2019; Tonitto et al., 2014; Whittinghill et al., 2012), but they mainly featured a simple, static decomposition retardation factor on lignin decomposition, without considering other possible responses such as decomposition stimulation and microbial physiological changes dynamically (Fierer et al., 2012; Gravuer and Eskelinen, 2017; Hu et al., 2022; Knorr et al., 2005; Sinsabaugh et al., 2016). Here, we aim to fill this gap by creating a dynamic model embodying multiple N-induced decomposer responses."

6. "L166: Was the turnover rate parameter for SOC3 adjusted based on simulations of the total SOC stock, or were measurements of MAOC stocks available? It would also be useful to mention the new value for the turnover rate parameter for SOC3 here. Is this value in line with reported turnover times of MAOC in forest topsoils?"

"L368: It would be good to report the value for the 'low SOC3 parameter' here"

"L485: 'enhanced SOC3 turnover': I would suggest to report this turnover rate and evaluate if this is in line with previously reported turnover rates of MAOC in forest soils."

We appreciate these suggestions about SOC3 turnover. We have reported the parameter values in **line 169 and line 387** and an additional supplementary figure (as **Fig. S4**) that reports the simulated turnover time ranges of MAOC in the ICP-II sites using the base model. Also, we did not do parameter "adjustment" strictly speaking, as we tested the default values from different versions of CENTURY (cf. **lines 167-171**).

Figure S4. The simulated C stocks and turnover times (C stock/ total C outflux) of the ICP-II sites using the selected base model.

We will not further adjust the SOC3 parameter in this manuscript, as the ranges of turnover times shown in Fig. S4 are not unreasonable, and because our study is not about parameter estimation. In general, parameter adjustment does not make sense in the absence of good data, and calibration with total C stock data is typically a bad practice due to model equifinality (see Van de Broek et al., 2025). Our Swiss data may also not be strictly representative of the meta-analysis sites (cf. answer #3 above). Using default parameter values is acknowledging that CENTURY model developers have better data (e.g., 14C, CO2 fluxes) for parameter estimations than we did. Nonetheless, we further discussed this limitation in **lines 487-490**:

"First, simulated C turnover rates may be too low, as the overestimation was greatly alleviated by an increase in SOC3 (MAOC) turnover. The simulated turnover times of MAOC ranged from 550 – 1500 y, which may be too high as other studies have reported turnover times as low as decades (Guo et al., 2022; Kleber et al., 2015; Lavallee et al., 2020). Increasing MAOC turnover also improved SOC:N ratios (which largely correlate with POC:MAOC ratios), implying this adjustment was in the right direction."

Furthermore, as shown in the **new Fig. 4d** (comparing with temperate forest response) and the long-term MAOC response in **Fig. S5**, readers can have a good understanding on how MAOC turnover rates can affect the match to the observed response.

7. "L187: It would be good to provide references for these meta-analyses."

We have referenced "See Table S5 in Supplement 2", which contains the correct references to both the numerical parameter values and the source meta-analyses.

8. "Section 2.3: I appreciate the authors testing many N responses. I would encourage the authors to provide graphs that show how the proposed variables and rate modifiers vary in response to the independent variables, as this is difficult to understand from the equations alone. In addition, it would

be good to mention if the proposed equations are based on literature, or if the authors formulated these equations themselves.

L244: how is the 'monthly priming effect' defined?"

The main text is already fairly over-crowded, and the equation graphs are available as **Fig. S10** in the current manuscript version. Unfortunately, we did forget to refer to this supplementary figure in the submitted manuscript, and so we have added its reference in **line 258** after the description of the equations: "The forms of these equations are presented graphically in Fig. S10 and described in greater detail in Supplement 2 sect. 5."

To improve the general understanding of the new equations (including the monthly priming effect), we have further provided in-depth equation descriptions and rationales in *Supplement 2 sect. 5*. Specifically, concerning the monthly priming effect, we added in *Supplement 2* lines 403-405:

"This explanatory variable "relative microbial biomass" is chosen in a way that a value of "1" results in no priming (neither positive nor negative priming). A value above one represents the "extent" of microbial biomass C in excess relative to the amount of C substrates."

**And in **lines 412-414**:**

"This non-linear form of function is chosen to approximate Michaelis-Menten kinetics, as exoenzymes and microbial biomass do decouple in reality (dormancy, competition, microbial cheaters, etc.), and enzymes do have a saturating effect especially when the concentration of enzymes greatly exceeds C substrates."

9. "L254-255: I suggest reconsidering the subscript 'retard"

Thanks for the suggestion, but we think that decomposition "retard-ation" is a sufficiently common terminology in soil science. Its meaning is quite clear given that "retard" and "retardation" were mentioned frequently throughout the manuscript.

10. "L259: It would be good to provide citations for these 'widespread experiments'"

This line is deleted and no longer exists in the revised manuscript, as we deemed it is redundant.

11. "L287 and L301: the correlation coefficient 'Pearson r' is not a measure for model performance, but for the proportion of explained variation. I would suggest reporting an error measure such as RMSE instead."

We agree and we have included proper error metrics alongside the correlation coefficients. to evaluate the N deposition model input data, in **lines 290-292**, we have added:

"The reliability of these map data was validated with the measured throughfall data from the aforementioned ICP Level II sites (Pearson r = 0.89, mean error =  $+3.2 \text{ kg N y}^{-1}$ . Note the systematic overestimation is because the raster map data represents total N deposition above the canopy, whereas ICP-II measurements are only throughfall N fluxes)."

To evaluate the model litter input data, we added in **line 306**: "We checked the validity of the simulated aboveground litter against plot-level measurements from the ICP Level II sites (Pearson r = 0.79, mean absolute error = 59.6 g C m-2 y-1)"

12. "Section 2.5: it would be good to mention here down to which depth the simulations were performed"

We have added in **line 312**: ""). In all simulations, we first spinned up the models for 3000 years until all C pools (both in the surface and 20 cm mineral soil layers) were at steady-state."

13. "L307: '[...] until the pools were at steady-state': for a better evaluation of the model performance by the reader, it would be good to report the simulated turnover times of the model pools, and how C was distributed between SOC1, SOC2 and SOC3."

"L354-358: It seems that the base model was not able to accurately simulate total SOC stocks for the Swiss forest sites (Fig. 3b). The authors justify this by stating that 'a close quantitative match of C stocks may not actually be desirable [...]'. However, I encourage the authors to better justify why they thrust results of a model that is not able to simulate total SOC stocks accurately, and to better explain why this was the case. For example, why were the model parameters not calibrated to match the conditions in Swiss forests? To evaluate the model results, an even more important aspect of the model outcomes is the simulated distribution of SOC and N over the different model pools, and the simulated turnover times of these pools (e.g., which portion of simulated SOC was present in SOC1, SOC2 and SOC3). I would encourage the authors to also report this information."

We appreciate the comments and we have included the distribution of turnover times and C stocks of SOC1, SOC2 and SOC3, reported in section 3.1 Base model selection and referred to as Fig. S4 (see answer #6). We think the turnover time ranges (especially MAOC), although at the slow end, are acceptable (cf. Brunmayr et al., 2024; Lavallee et al., 2020), and we will not further adjust the turnover parameters. As mentioned in answer #6, parameter calibration requires good data, and calibrating to fit total SOC stocks is typically a bad practice due to equifinality (Van de Broek et al., 2025).

Concerning the overestimation of the total SOC stock (to a lesser extent LFH stock), we reasoned that if fully accurate historical input data for running simulations is not available, SOC stock should not match with reality (lines 362-365). Expecting (or calibrating to) a precise match of SOC stocks under this uncertainty of historical inputs implies the model is simulating the C stocks correctly for the wrong reasons, which may bias future projections (our N addition simulations can be considered as future projections). Our main justifications for the mismatch (overestimation) were in lines 486-493 and lines 498-502 i.e., past management likely reduced real-world SOC stocks and that the CENTURY's pH retardation effect likely inflated simulated SOC stocks. We think these are sufficiently reasonable explanations. To aid in the understanding of how these overestimated C stocks may affect our results or conclusions, we have added the following information:

**In lines 398-399:**

"It must be noted however that LFH response was strongly reduced by overestimated initial LFH stocks in the Nexretard + Nlimstim models (Table 2 & Fig. 3)."

**And in **lines 437-438**":**

"...Lastly, the percentage responses of LFH, SOC and POC were reduced by larger (i.e., overestimated) initial C stocks (cf. Fig. 3), also only in the Nexretard + Nlimstim models."

14. "Fig. 5a: while the regression line looks fine, a large portion of the simulated data is overestimated (y-values between 13-16), while another portion is underestimated (y-values between 7-9). This does not seem in line with the statement of 'the models capturing MAOC:N reasonably'. I would suggest the authors to revise this statement."

We agree and have revised lines 447-448:

"The models captured the general trend of MAOC:N but there was a cluster of underestimations associated with broadleaved forests ( $R^2 = 0.33$ , P < 0.001, Fig. 5a). However, it matched the observed correlations with environmental factors fairly (Fig. 5c)."

15. "L509: what is meant by 'simplistic' decomposition?"

We have changed "simplistic decomposition retardation" to "decomposition retardation alone" (**line** 531 in the revised manuscript)

16. "Fig. 1: It was not clear to me what the y-axis label 'Labile C:N' means. Can this be clarified?"

We have further explained these terms in the Fig. 1 caption:

"Conceptual model of the hypothesized relationship of microbially-available resource C:N (labile C:N), microbial resource C:N demand (TER), and decomposition rate. ..."

17. "Table 1: In the second row under (2.), should it be 'increasing N limitation?', instead of decreasing?"

Decreasing is correct. However, we changed it to "Decomposition stimulation with driven by decreasing N-limitation" for clarity.

18. "Fig. 3: a unit needs to be reported for the RMSE"

RMSE is in the same unit as the dependent variable, which is already mentioned in Fig. 3 (y-axis labels). The ratios have no unit.

19. "Fig. 7: it would be good to provide more information about this figure in the caption."

We agree and have expanded the Fig.7 caption as: "A synoptic overview of the main decomposition effects of boundary N inputs on various organic matter pools. The illustrated effect sizes depict the general trends concluded from this study, but it must be noted that the effect varies for individual soils and organic matter that has a wide range of C:N ratios."

**References (only new references not in the current manuscript text)**

Brunmayr, A. S., Hagedorn, F., Moreno Duborgel, M., Minich, L. I., and Graven, H. D.: Radiocarbon analysis reveals underestimation of soil organic carbon persistence in new-generation soil models, Geoscientific Model Development, 17, 5961–5985, https://doi.org/10.5194/gmd-17-5961-2024, 2024.

- Buchkowski, R. W., Schmitz, O. J., and Bradford, M. A.: Microbial stoichiometry overrides biomass as a regulator of soil carbon and nitrogen cycling, Ecology, 96, 1139–1149, https://doi.org/10.1890/14-1327.1, 2015.
- Lavallee, J. M., Soong, J. L., and Cotrufo, M. F.: Conceptualizing soil organic matter into particulate and mineral-associated forms to address global change in the 21st century, Global Change Biology, 26, 261–273, https://doi.org/10.1111/gcb.14859, 2020.
- Luo, Y., Weng, E., Wu, X., Gao, C., Zhou, X., and Zhang, L.: Parameter Identifiability, Constraint, and Equifinality in Data Assimilation with Ecosystem Models, Ecological Applications, 19, 571–574, 2009.
- Marschmann, G. L., Pagel, H., Kügler, P., and Streck, T.: Equifinality, sloppiness, and emergent structures of mechanistic soil biogeochemical models, Environmental Modelling & Software, 122, 104518, https://doi.org/10.1016/j.envsoft.2019.104518, 2019.
- Rocci, K. S., Cleveland, C. C., Eastman, B. A., Georgiou, K., Grandy, A. S., Hartman, M. D., Hauser, E., Holland-Moritz, H., Kyker-Snowman, E., Pierson, D., Reich, P. B., Schlerman, E. P., and Wieder, W. R.: Aligning theoretical and empirical representations of soil carbon-to-nitrogen stoichiometry with process-based terrestrial biogeochemistry models, Soil Biology and Biochemistry, 189, 109272, https://doi.org/10.1016/j.soilbio.2023.109272, 2024.
- Schimel, J. P. and Weintraub, M. N.: The implications of exoenzyme activity on microbial carbon and nitrogen limitation in soil: a theoretical model, Soil Biology and Biochemistry, 35, 549–563, https://doi.org/10.1016/S0038-0717(03)00015-4, 2003.
- Van de Broek, M., Govers, G., Schrumpf, M., and Six, J.: A microbially driven and depth-explicit soil organic carbon model constrained by carbon isotopes to reduce parameter equifinality, Biogeosciences, 22, 1427–1446, https://doi.org/10.5194/bg-22-1427-2025, 2025.
- Woodward, F. I., Lomas, M. R., and Kelly, C. K.: Global climate and the distribution of plant biomes, Philosophical Transactions of the Royal Society of London. Series B: Biological Sciences, 359, 1465–1476, https://doi.org/10.1098/rstb.2004.1525, 2004.

Zechmeister-Boltenstern, S., Michel, K., and Pfeffer, M.: Soil microbial community structure in European forests in relation to forest type and atmospheric nitrogen deposition, Plant Soil, 343, 37–50, https://doi.org/10.1007/s11104-010-0528-6, 2011.

---

## Author Response (AR2)

In this round of review, we only received minor comments, and we made all the changes accordingly:

- 1. We proofread the entire manuscript and run it through grammar checkers (corrected roughly 20 mistakes) and improved the overall English delivery.
- 2. We increased the font size of the axis labels in Figures 3, 4 and 5. We also checked and improved the quality of all the figures and legends.
- 3. We consolidated the Abstract into one single paragraph.
- 4. We added a key figure as per the submission guideline.
- 5. We also removed one supplementary figure (Fig. S2 in the previous version) in Supplement 1 as we deemed it as unnecessary.